# Score-based Generative Neural Networks for Large-Scale Optimal Transport

**Max Daniels**
Northeastern University
daniels.g@northeastern.edu

**Tyler Maunu** *
Brandeis University
maunu@brandeis.edu

**Paul Hand**
Northeastern University
p.hand@northeastern.edu

## Abstract

We consider the fundamental problem of sampling the optimal transport coupling between given source and target distributions. In certain cases, the optimal transport plan takes the form of a one-to-one mapping from the source support to the target support, but learning or even approximating such a map is computationally challenging for large and high-dimensional datasets due to the high cost of linear programming routines and an intrinsic curse of dimensionality. We study instead the Sinkhorn problem, a regularized form of optimal transport whose solutions are couplings between the source and the target distribution. We introduce a novel framework for learning the Sinkhorn coupling between two distributions in the form of a score-based generative model. Conditioned on source data, our procedure iterates Langevin Dynamics to sample target data according to the regularized optimal coupling. Key to this approach is a neural network parametrization of the Sinkhorn problem, and we prove convergence of gradient descent with respect to network parameters in this formulation. We demonstrate its empirical success on a variety of large scale optimal transport tasks.

## 1 Introduction

It is often useful to compare two data distributions by computing a distance between them in some appropriate metric. For instance, statistical distances can be used to fit the parameters of a distribution to match some given data. Comparison of statistical distances can also enable distribution testing, quantification of distribution shifts, and provide methods to correct for distribution shift through domain adaptation [12].

Optimal transport theory provides a rich set of tools for comparing distributions in *Wasserstein Distance*. Intuitively, an optimal transport plan from a source distribution $\sigma \in \mathcal{M}_+(\mathcal{X})$ to a target distribution $\tau \in \mathcal{M}_+(\mathcal{Y})$ is a blueprint for transporting the mass of $\sigma$ to match that of $\tau$ as cheaply as possible with respect to some ground cost. Here, $\mathcal{X}$ and $\mathcal{Y}$ are compact metric spaces and $\mathcal{M}_+(\mathcal{X})$ denotes the set of positive Radon measures over $\mathcal{X}$, and it is assumed that $\sigma, \tau$ are supported over all of $\mathcal{X}, \mathcal{Y}$ respectively. The Wasserstein Distance between two distributions is defined to be the cost of an optimal transport plan.

Because the ground cost can incorporate underlying geometry of the data space, optimal transport plans often provide a meaningful correspondence between points in $\mathcal{X}$ and $\mathcal{Y}$. A famous example is given by Brenier's Theorem, which states that, when $\mathcal{X}, \mathcal{Y} \subseteq \mathbb{R}^d$ and $\sigma, \tau$ have finite variance, the optimal transport plan under a squared-$l_2$ ground cost is realized by a map $T : \mathcal{X} \to \mathcal{Y}$ [26, Theorem 2.12]. However, it is often computationally challenging to exactly compute optimal transport plans, as one must exactly solve a linear program requiring time which is super-quadratic in the size of input datasets [5].

---

*Work done while an Instructor in Applied Mathematics at MIT.

35th Conference on Neural Information Processing Systems (NeurIPS 2021).

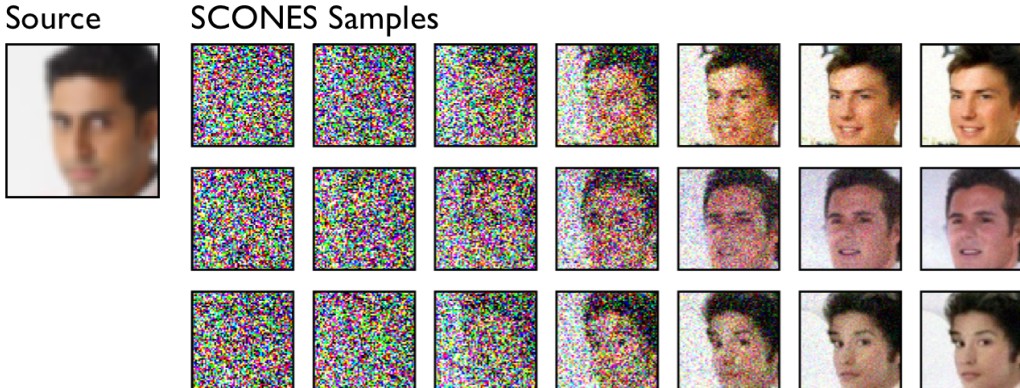

Figure 1: We use SCONES to sample the mean-squared-$L^2$ cost, entropy regularized optimal transport mapping between 2x downsampled CelebA images (Source) and unmodified CelebA images (Target) at $\lambda = 0.005$ regularization.

Instead, we opt to study a regularized form of the optimal transport problem whose solution takes the form of a joint density $\pi(x, y)$ with marginals $\pi_X(x) = \sigma(x)$ and $\pi_Y(y) = \tau(y)$. A correspondence between points is given by the conditional distribution $\pi_{Y|X=x}(y)$, which relates each input point to a distribution over output points.

In recent work [22], the authors propose a large-scale stochastic dual approach in which $\pi(x, y)$ is parametrized by two continuous dual variables that may be represented by neural networks and trained at large-scale via stochastic gradient ascent. Then, with access to $\pi(x, y)$, they approximate an optimal transport *map* using a barycentric projection of the form $T : x \mapsto \arg\min_y \mathbb{E}_{\pi_{Y|X=x}}[d(y, Y)]$, where $d : \mathcal{Y} \times \mathcal{Y} \to \mathbb{R}$ is a convex cost on $\mathcal{Y}$. Their method is extended by [15] to the problem of learning regularized Wasserstein barycenters. In both cases, the Barycentric projection is observed to induce averaging artifacts such as those shown in Figure 2.

Instead, we propose a direct sampling strategy to generate samples from $\pi_{Y|X=x}(y)$ using a *score-based generative model*. Score-based generative models are trained to sample a generic probability density by iterating a stochastic dynamical system knows as *Langevin dynamics* [24]. In contrast to projection methods for large-scale optimal transport, we demonstrate that pre-trained score based generative models can be naturally applied to the problem of large-scale regularized optimal transport. Our main contributions are as follows:

1. We show that pretrained score based generative models can be easily adapted for the purpose of sampling high dimensional regularized optimal transport plans. Our method eliminates the need to estimate a barycentric projection and it results in sharper samples because it eliminates averaging artifacts incurred by such a projection.

2. Score based generative models have been used for unconditional data generation and for conditional data generation in settings such as inpainting. We demonstrate how to adapt pretrained score based generative models for the more challenging conditional sampling problem of *regularized optimal transport*.

3. Our method relies on a neural network parametrization of the dual regularized optimal transport problem. Under assumptions of large network width, we prove that gradient descent w.r.t. neural network parameters converges to a global maximizer of the dual problem. We also prove optimization error bounds based on a stability analysis of the dual problem.

4. We demonstrate the empirical success of our method on a synthetic optimal transport task and on optimal transport of high dimensional image data.

## 2   Background and Related Work

We will briefly review some key facts about optimal transport and generative modeling. For a more expansive background on optimal transport, we recommend the references [26] and [25].

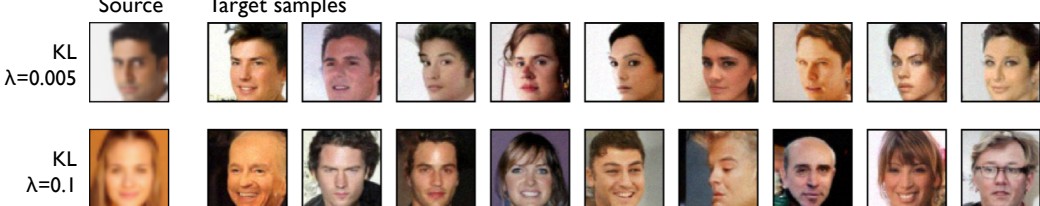

Figure 2: Samples generated by SCONES for entropy regularized optimal transport including the samples shown in Figure 1. At regularization $\lambda = 0.005$, optimal transportation with $L^2$ cost has a visible effect on generated images. This effect diminishes at increased regularization $\lambda = 0.1$.

## 2.1 Regularized Optimal Transport

We begin by reviewing the formulation of the regularized OT problem.

**Definition 2.1** (Regularized OT). Let $\sigma \in \mathcal{M}_+(\mathcal{X})$ and $\tau \in \mathcal{M}_+(\mathcal{Y})$ be probability measures supported on compact sets $\mathcal{X}, \mathcal{Y}$. Let $c : \mathcal{X} \times \mathcal{Y} \to \mathbb{R}$ be a convex, lower semi-continuous function representing cost of transporting a point $x \in \mathcal{X}$ to $y \in \mathcal{Y}$. The regularized optimal transport distance $\mathrm{OT}_\lambda(\sigma, \tau)$ is given by

$$\mathrm{OT}_\lambda(\sigma, \tau) = \min_\pi \mathbb{E}_\pi[c(x, y)] + \lambda H(\pi) \tag{1}$$
$$\text{subject to} \quad \pi_X = \sigma, \quad \pi_Y = \tau$$
$$\pi(x, y) \geq 0$$

where $H : \mathcal{M}_+(\mathcal{X} \times \mathcal{Y}) \to \mathbb{R}$ is a convex regularizer and $\lambda \geq 0$ is a regularization parameter.

We are mainly concerned with optimal transport of empirical distributions, where $\mathcal{X}$ and $\mathcal{Y}$ are finite and $\sigma, \tau$ are empirical probability vectors. In most of the following theorems, we will work in the *empirical setting* of Definition 2.1, so that $\mathcal{X}$ and $\mathcal{Y}$ are finite subsets of $\mathbb{R}^d$ and $\sigma, \tau$ are vectors in the probability simplices of dimension $|\mathcal{X}|$ and $|\mathcal{Y}|$, respectively.

We refer to the objective $K_\lambda(\pi) = \mathbb{E}_\pi[c(x, y)] + \lambda H(\pi)$ as the *primal objective*, and we will use $J_\lambda(\varphi, \psi)$ to refer to the associated *dual objective*, with dual variables $\varphi, \psi$. Two common regularizers are $H(\pi) = \mathrm{KL}(\pi || \sigma \times \tau)$ and $H(\pi) = \chi^2(\pi || \sigma \times \tau)$, sometimes called *entropy* and $l_2$ regularization respectively:

$$\mathrm{KL}(\pi || \sigma \times \tau) = \mathbb{E}_\pi \left[ \log \left( \frac{d\pi(x, y)}{d\sigma(x) d\tau(y)} \right) \right], \quad \chi^2(\pi || \sigma \times \tau) = \mathbb{E}_{\sigma \times \tau} \left[ \left( \frac{d\pi(x, y)}{d\sigma(x) d\tau(y)} \right)^2 \right]$$

where $\frac{d\pi(x,y)}{d\sigma(x) d\tau(y)}$ is the Radon-Nikodym derivative of $\pi$ with respect to the product measure $\sigma \times \tau$. These regularizers contribute useful optimization properties to the primal and dual problems.

For example, $\mathrm{KL}(\pi || \sigma \times \tau)$ is exactly the mutual information $I_\pi(X; Y)$ of the coupling $(X, Y) \sim \pi$, so intuitively speaking, entropy regularization explicitly prevents $\pi_{Y|X=x}$ from concentrating on a point by stipulating that the conditional measure retain some bits of uncertainty after conditioning. The effects of this regularization are described by Propositions 2.2 and 2.3.

First, regularization induces convexity properties which are useful from an optimization perspective.

**Proposition 2.2.** *In the empirical setting of Definition 2.1, the entropy regularized primal problem $K_\lambda(\pi)$ is $\lambda$-strongly convex in $l_1$ norm. The dual problem $J_\lambda(\varphi, \psi)$ is concave, unconstrained, and $\frac{1}{\lambda}$-strongly smooth in $l_\infty$ norm. Additionally, these objectives witness strong duality: $\inf_{\pi \in \mathcal{M}_+(\mathcal{X} \times \mathcal{Y})} K_\lambda(\pi) = \sup_{\varphi, \psi \in \mathbb{R}^{2d}} J_\lambda(\varphi, \psi)$, and the extrema of each objective are attained over their respective domains.*

In addition to these optimization properties, regularizing the OT problem induces a specific form of the dual objective and resulting optimal solutions.

**Proposition 2.3.** *In the setting of Proposition 2.2, the KL-regularized dual objective takes the form*

$$J_\lambda(\varphi, \psi) := \mathbb{E}_\sigma[\varphi(x)] + \mathbb{E}_\tau[\psi(y)]$$
$$- \lambda \mathbb{E}_{\sigma \times \tau} \left[ \frac{1}{e} \exp \left( \frac{1}{\lambda} (\varphi(x) + \psi(y) - c(x, y)) \right) \right].$$

*The optimal solutions $\varphi^*, \psi^* = \arg\max_{\varphi, \psi \in \mathbb{R}^{2d}} J_\lambda(\varphi, \psi)$ and $\pi^* = \arg\min_{\pi \in \mathcal{M}_+(\mathcal{X} \times \mathcal{Y})} K_\lambda(\pi)$ satisfy*

$$\pi^*(x, y) = \frac{1}{e} \exp \left( \frac{1}{\lambda} (\varphi^*(x) + \psi^*(y) - c(x, y)) \right) \sigma(x) \tau(y).$$

These propositions are specializations of Proposition 2.4 and they are well-known to the literature on entropy regularized optimal transport [5, 2]. The solution $\pi^*(x, y)$ of the entropy regularized problem is often called the *Sinkhorn coupling* between $\sigma$ and $\tau$ in reference to Sinkhorn's Algorithm [23], a popular approach to efficiently solving the discrete entropy regularized OT problem. For *arbitrary* choices of regularization, we call $\pi^*(x, y)$ a Sinkhorn coupling.

Propositions 2.2 and 2.3 illustrate the main desiderata when choosing the regularizer: that $H(\pi)$, and hence $K_\lambda(\pi)$, be strongly convex in $l_1$ norm and that $H(\pi)$ induce a nice analytic form of $\pi^*$ in terms of $\varphi^*, \psi^*$. In regards to the former, $H(\pi)$ is akin to barrier functions used by interior point methods [5]. Prior work [2] is an example of the latter, in which it is shown that for discrete optimal transport, $\chi^2$ regularization yields an analytic form of $\pi^*$ having a thresholding operation that promotes sparsity.

Conveniently, the KL and $\chi^2$ regularizers both belong to the class of $f$-Divergences, which are statistical divergences of the form

$$D_f(p||q) = \mathbb{E}_q \left[ f \left( \frac{p(x)}{q(x)} \right) \right].$$

where $f : \mathbb{R} \to \mathbb{R}$ is convex, $f(1) = 0$, $p, q$ are probability measures, and $p$ is absolutely continuous with respect to $q$. For example, the KL regularizer has $f_{\text{KL}}(t) = t \log(t)$ and the $\chi^2$ regularizer has $f_{\chi^2}(t) = t^2 - 1$. The $f$-Divergences are good choices for regularizing optimal transport: strong convexity of $f$ is a sufficient condition for strong convexity of $H_f(\pi) := D_f(\pi||\sigma \times \tau)$ in $l_1$ norm, and the form of $f$ is the aspect which determines the form of $\pi^*$ in terms of $\varphi^*, \psi^*$. This relationship is captured by the following generalization of Propositions 2.2 and 2.3, which we prove in Section B of the Supplemental Materials.

**Proposition 2.4.** *Consider the empirical setting of Definition 2.1. Let $f(v) : \mathbb{R} \to \mathbb{R}$ be a differentiable $\alpha$-strongly convex function with convex conjugate $f^*(v)$. Set $f^{*\prime}(v) = \partial_v f^*(v)$. Define the violation function $V(x, y; \varphi, \psi) = \varphi(x) + \psi(y) - c(x, y)$. Then,*

1. *The $D_f$ regularized primal problem $K_\lambda(\pi)$ is $\lambda\alpha$-strongly convex in $l_1$ norm. With respect to dual variables $\varphi \in \mathbb{R}^{|\mathcal{X}|}$ and $\psi \in \mathbb{R}^{|\mathcal{Y}|}$, the dual problem $J_\lambda(\varphi, \psi)$ is concave, unconstrained, and $\frac{1}{\lambda\alpha}$-strongly smooth in $l_\infty$ norm. Strong duality holds: $K_\lambda(\pi) \geq J_\lambda(\varphi, \psi)$ for all $\pi$, $\varphi$, $\psi$, with equality for some triple $\pi^*, \varphi^*, \psi^*$.*

2. *$J_\lambda(\varphi, \psi)$ takes the form $J_\lambda(\varphi, \psi) = \mathbb{E}_\sigma[\varphi(x)] + \mathbb{E}_\tau[\psi(y)] - \mathbb{E}_{\sigma \times \tau}[H_f^*(V(x, y; \varphi, \psi))]$, where $H_f^*(v) = \lambda f^*(\lambda^{-1} v)$.*

3. *The optimal solutions $(\pi^*, \varphi^*, \psi^*)$ satisfy*

$$\pi^*(x, y) = M_f(V(x, y; \varphi, \psi)) \sigma(x) \tau(y)$$

*where $M_f(x, y) = f^{*\prime}(\lambda^{-1} v)$.*

For this reason, we focus in this work on $f$-Divergence-based regularizers. Where it is clear, we will drop subscripts on regularizer $H(\pi)$ and the so-called *compatibility function* $M(v)$ and we will omit the dual variable arguments of $V(x, y)$. The specific form of these terms for KL regularization, $\chi^2$ regularization, and a variety of other regularizers may be found in Section A of the Appendix.

## 2.2 Langevin Sampling and Score Based Generative Modeling

Given access to optimal dual variables $\varphi^*(x)$, $\psi^*(y)$, it is easy to evaluate the density of the corresponding optimal coupling according to Proposition 2.4. To generate samples distributed according to this coupling, we apply *Langevin Sampling*. The key quantity used in Langevin sampling of a generic (possibly unnormalized) probability measure $p(x)$ is its *score function*, given by $\nabla_x \log p(x)$ for $x \in \mathcal{X}$. The algorithm is an iterative Monte Carlo method which generates approximate samples $\tilde{x}_t$ by iterating the map

$$\tilde{x}_t = \tilde{x}_{t-1} + \epsilon \nabla_x \log p(\tilde{x}_{t-1}) + \sqrt{2\epsilon} z_t$$

where $\epsilon > 0$ is a step size parameter and where $z_t \sim \mathcal{N}(0, I)$ independently at each time step $t \geq 0$. In the limit $\epsilon \to 0$ and $T \to \infty$, the samples $\tilde{x}_T$ converge weakly in distribution to $p(x)$. Song and Ermon [24] introduce a method to estimate the score with a neural network $s_\vartheta(x)$, trained on samples from $p(x)$, so that it approximates $s_\vartheta(x) \approx \nabla_x \log p(x)$ for a given $x \in \mathcal{X}$. To generate samples, one may iterate Langevin dynamics with the score estimate in place of the true score.

To scale this method to high dimensional image datasets, Song and Ermon [24] propose an annealing scheme which samples noised versions of $p(x)$ as the noise is gradually reduced. One first samples a noised distribution $p(x) * \mathcal{N}(0, \tau_1)$, at noise level $\tau_1$. The noisy samples, which are presumed to lie near high density regions of $p(x)$, are used to initialize additional rounds of Langevin dynamics at diminishing noise levels $\tau_2 > \ldots > \tau_N > 0$. At the final round, Annealed Langevin Sampling outputs approximate samples according to the noiseless distribution. Song and Ermon [24] demonstrate that Annealed Langevin Sampling (ALS) with score estimatation can be used to generate sample images that rival the quality of popular generative modeling tools like GANs or VAEs.

## 3 Conditional Sampling of Regularized Optimal Transport Plans

Our approach can be split into two main steps. First, we approximate the density of the optimal Sinkhorn coupling $\pi^*(x, y)$ which minimizes $K_\lambda(\pi)$ over the data. To do so, we apply the large-scale stochastic dual approach introduced by Seguy et al. [22], which involves instantiating neural networks $\varphi_\theta : \mathcal{X} \to \mathbb{R}$ and $\psi_\theta : \mathcal{Y} \to \mathbb{R}$ that serve as parametrized dual variables. We then maximize $J_\lambda(\varphi_\theta, \psi_\theta)$ with respect to $\theta$ via gradient descent and take the resulting parameters $\theta^*$ and the associated transport plan $\hat{\pi}(x, y) = M(V(x, y; \varphi_{\theta^*}, \psi_{\theta^*}))\sigma(x)\tau(y)$. This procedure is shown in Algorithm 1. Note that when the dual problem is only approximately maximized, $\hat{\pi}$ need not be a normalized density. We therefore call $\tilde{\pi}$ the *pseudo-coupling* which approximates the true Sinkhorn coupling $\hat{\pi}$.

After optimizing $\theta^*$, we sample the conditional $\hat{\pi}_{Y|X=x}(y)$ using Langevin dynamics. The score estimator for the conditional distribution is,

$$\nabla_y \log \hat{\pi}_{Y|X=x}(y) = \nabla_y [\log(M(V(x, y; \varphi_{\theta^*}, \psi_{\theta^*}))\sigma(x)\tau(y)) - \log(\sigma(x))]$$
$$\approx \nabla_y \log(M(V(x, y; \varphi_{\theta^*}, \psi_{\theta^*}))) + s_\vartheta(y).$$

We therefore approximate $\nabla_y \hat{\pi}_{Y|X=x}(y)$ by directly differentiating $\log M(V(x, y))$ using standard automatic differentiation tools and adding the result to an unconditional score estimate $s_\vartheta(y)$. The full Langevin sampling algorithm for general regularized optimal transport is shown in Algorithm 2.

We note that our method has the effect of biasing the Langevin iterates towards the region where $\hat{\pi}_{Y|X=x}$ is localized. This may be beneficial for Langevin sampling, which enjoys exponentially fast mixing when sampling log-concave distributions. In the supplementary material, we prove a known result that for the entropy regularized problem given in Proposition 2.3: the compatibility $M(V(x, y; \varphi^*, \psi^*)) = \frac{1}{e} \exp\left(\frac{1}{\lambda}\left(\varphi^*(x) + \psi^*(y) - c(x, y)\right)\right)$ is log-concave with respect to $y$. For $\lambda \to 0$, this localizes around the optimal transport of $x$, $T(x)$, and so heuristically should lead to faster mixing.

## 4 Theoretical Analysis

In principle, Definition 2.1 poses an unconstrained optimization problem over $\mathbb{R}^{2d}$, which could be optimized directly by gradient descent on $\varphi$, $\psi$ as vectors. The point of a more expensive neural network parametrization $\varphi_\theta$, $\psi_\theta$ is to learn a *continuous* distribution that agrees with the empirical

| **Algorithm 1** Density Estimation. | **Algorithm 2** SCONES Sampling Procedure |
|---|---|
| **Input**: Step size $\gamma$, batch size $m$ | **Input**: Noise levels $\tau_1 > \ldots > \tau_N$. |
| **Input**: Nets $\varphi_{\theta_1}, \psi_{\theta_2}$. | **Input**: Dual vars. $\tilde{\varphi}(x), \tilde{\psi}(y)$. Source $x \in \mathcal{X}$. |
| **Input**: Datasets $\sigma, \tau$. Time steps $T > 0$. | **Input**: Time steps $T > 0$. Step size $\epsilon > 0$. |
| **Output**: Trained $\varphi_{\theta_1^*}, \psi_{\theta_2^*}$. | **Output**: Data sample $\tilde{y} \sim \pi_{Y|X=x}(y)$. |
| **for** $t = 1 \ldots T$. **do** | Initialize $\tilde{y}_{1,0} \sim \mathcal{N}(0, I)$. |
| $\quad$ Sample $X_1, \ldots, X_m \sim \sigma$, | **for** $\tau_i, i = 1 \ldots N$ **do** |
| $\quad$ and $Y_1, \ldots, Y_m \sim \tau$. | $\quad$ **for** $t = 1 \ldots T$. **do** |
| $\quad$ Stochastic gradient update $\varphi_{\theta_1}, \psi_{\theta_2}$: | $\quad\quad$ Sample $z \sim \mathcal{N}(0, \tau_i)$. |
| $\quad \Delta_1 \leftarrow \sum\limits_{i,j=1}^{m} \nabla_{\theta_1}\left[\varphi_{\theta_1}(X_i) - H^*(V(X_i, Y_j))\right].$ | $\quad\quad$ Compute score update: |
| | $\quad\quad \Delta_s \leftarrow s_\vartheta(\tilde{y}_{i,t-1}).$ |
| $\quad \Delta_2 \leftarrow \sum\limits_{i,j=1}^{m} \nabla_{\theta_2}\left[\psi_{\theta_2}(Y_j) - H^*(V(X_i, Y_j))\right].$ | $\quad\quad \Delta_\pi \leftarrow \nabla_y \log M(V(x, \tilde{y}_{i,t-1}; \tilde{\varphi}, \tilde{\psi})).$ |
| | $\quad\quad \tilde{x}_{i,t} = \tilde{x}_{i,t-1} + (\epsilon/2)(\Delta_s + \Delta_\pi) + \sqrt{\epsilon}z.$ |
| $\quad \theta_1 \leftarrow \theta_1 + \gamma \Delta_1.$ | $\quad$ **end for** |
| $\quad \theta_2 \leftarrow \theta_2 + \gamma \Delta_2.$ | $\quad$ Initialize $\tilde{x}_{i+1,0} = \tilde{x}_{i,T}.$ |
| **end for** | **end for** |
| Output parameters $\{\theta_1, \theta_2\}$. | Output sample $\tilde{x}_{N,T}.$ |

optimal transport plan between discrete $\sigma, \tau$, and that generalizes to a continuous space containing $\mathcal{X} \times \mathcal{Y}$. By training $\varphi_\theta, \psi_\theta$, we approximate the underlying continuous data distribution up to optimization error and up to statistical estimation error between the empirical coupling and the population coupling. In the present section, we justify this approach by proving convergence of Algorithm 1 to the global maximizer of $J_\lambda(\varphi, \psi)$, under assumptions of large network width, along with a quantitative bound on optimization error. In Section B.1 of the Appendix, we provide a cursory analysis of rates of statistical estimation of entropy regularized Sinkhorn couplings. We make the following main assumptions on the neural networks $\varphi_\theta$ and $\psi_\theta$.

**Assumption 4.1** (Approximate Linearity). Let $f_\theta(x)$ be a neural network and set $\mathcal{K}_\theta(x) = [J_\theta^f(x)][J_\theta^f(x)]^T$ where $J_\theta^f(x)$ is the Jacobian of $f_\theta(x)$ with respect to $\theta$. Let $\Theta$ be a set of feasible weights, for example those reachable by gradient descent. Then $f_\theta(x)$ must satisfy,

1. There exists $R \gg 0$ so that $\Theta \subseteq B(0, R)$, where $B(0, R)$ is the Euclidean ball of radius $R$.

2. There exist $\rho_M > \rho_m > 0$ such that for $\theta \in \Theta$ and for all data points $\{X_i\}_{i=1}^N$,

$$\rho_M \geq \lambda_{\max}(\mathcal{K}_\theta(X_i)) \geq \lambda_{\min}(\mathcal{K}_\theta(X_i)) \geq \rho_m > 0.$$

3. For $\theta \in \Theta$ and for all data points $\{X_i\}_{i=1}^N$, the Hessian matrix $D_\theta^2 f_\theta(x_i)$ is bounded in spectral norm: $\|D_\theta^2 f_\theta(x_i)\| \leq \frac{\rho_M}{C_h}$, where $C_h \gg 0$ depends only on $R$, $N$, and the regularization $\lambda$.

The dependencies of $C_h$ are made clear in the Supplemental Materials, Section B. The quantity $\mathcal{K}_\theta(x)$ is called the *neural tangent kernel* (NTK) associated with the network $f_\theta(x)$. It has been shown for a variety of nets that, at sufficiently large width, the NTK is well conditioned and nearly constant on the set of weights reachable by gradient descent on a convex objective function [16, 6, 7]. For instance, fully-connected networks with smooth and Lipschitz-continuous activations fall into this class and hence satisfy Assumption 4.1 when the width of all layers is sufficiently large [16].

First, we show in Theorem 4.2 that when $\varphi, \psi$ are parametrized by neural networks satisfying Assumption 4.1, gradient descent converges to *global* maximizers of the dual objective. This provides additional justification for the large-scale approach of Seguy et al. [22].

**Theorem 4.2** (Optimizing Neural Nets). *Suppose $J_\lambda(\varphi, \psi)$ is $\frac{1}{s}$-strongly smooth in $l_\infty$ norm. Let $\varphi_\theta$, $\psi_\theta$ be neural networks satisfying Assumption 4.1 for the dataset $\{(x_i, y_i)\}_{i=1}^N$, $N = |\mathcal{X}| \cdot |\mathcal{Y}|$.*

*Then gradient descent of $J_\lambda(\varphi_\theta, \psi_\theta)$ with respect to $\theta$ at learning rate $\eta = \frac{\lambda}{2\rho_M}$ converges to an $\epsilon$-approximate global maximizer of $J_\lambda$ in at most $\left(\frac{2\kappa R^2}{s}\right)\epsilon^{-1}$ iterations, where $\kappa = \frac{\rho_M}{\rho_m}$.*

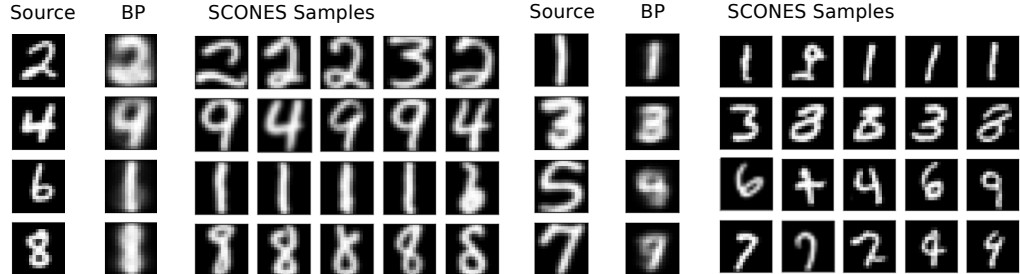

Figure 3: Comparison of Barycentric Projection [22] to SCONES for optimal transport between USPS and MNIST datasets of handwritten digits. (Left) Transporting MNIST to USPS. (Right) Transporting USPS to MNIST. Here, we show transportation of the $\chi^2$ regularized problem at $\lambda = 0.001$.

Given outputs $\hat{\varphi}$, $\hat{\psi}$ of Algorithm 1, we may assume by Theorem 4.2 that the networks are $\epsilon$-approximate global maximizers of $J_\lambda(\varphi, \psi)$. Due to $\lambda\alpha$-strong convexity of the primal objective, the optimization error $\epsilon$ bounds the distance of the underlying pseudo-plan $\hat{\pi}$ from the true global extrema. We make this bound concrete in Theorem 4.3, which guarantees that approximately maximizing $J_\lambda(\varphi, \psi)$ is sufficient to produce a close approximation of the true empirical Sinkhorn coupling.

**Theorem 4.3** (Stability of the OT Problem). *Suppose $K_\lambda(\pi)$ is $s$-strongly convex in $l_1$ norm and let $\mathcal{L}(\varphi, \psi, \pi)$ be the Lagrangian of the regularized optimal transport problem. For $\hat{\varphi}$, $\hat{\psi}$ which are $\epsilon$-approximate maximizers of $J_\lambda(\varphi, \psi)$, the pseudo-plan $\hat{\pi} = M_f(V(x, y; \hat{\varphi}, \hat{\psi}))\sigma(x)\tau(y)$ satisfies*

$$|\hat{\pi} - \pi^*|_1 \leq \sqrt{\frac{2\epsilon}{s}} \leq \frac{1}{s}\left|\nabla_{\hat{\pi}}\mathcal{L}(\hat{\varphi}, \hat{\psi}, \hat{\pi})\right|_1.$$

Theorem 4.3 guarantees that if one can approximately optimize the dual objective using Algorithm 1, then the corresponding coupling $\hat{\pi}$ is close in $l_1$ norm to the true optimal transport coupling. This approximation guarantee justifies the choice to draw samples $\tilde{\pi}_{Y|X=x}$ as an approximation to sampling $\pi_{Y|X=x}$ instead. Both Theorems 4.2 and 4.3 are proven in Section B of the Appendix.

## 5 Experiments

Our main point of comparison is the barycentric projection method proposed by Seguy et al. [22], which trains a neural network $T_\theta : \mathcal{X} \to \mathcal{Y}$ to map source data to target data by optimizing the objective $\theta := \arg\min_\theta \mathbb{E}_{\pi_{Y|X=x}}[|T_\theta(x) - Y|^2]$. For transportation experiments between USPS [19] and MNIST [13] datasets, we scale both datasets to 16px and parametrize the dual variables and barycentric projections by fully connected ReLU networks. We train score estimators for MNIST and USPS at 16px resolution using the method and architecture of [24]. For transportation experiments using CelebA [17], dual variables $\varphi$, $\psi$ are parametrized as ReLU FCNs with 8, 2048-dimensional hidden layers. Both the barycentric projection and the score estimators use the U-Net based image-to-image architecture introduced in [24]. Numerical hyperparameters like learning rates, optimizer parameters, and annealing schedules, along with additional details of our neural network architectures, are tabulated in Section C of the Appendix.

### 5.1 Optimal Transportation of Image Data

We show in Figure 3 a qualitative plot of SCONES samples on transportation between MNIST and USPS digits. We also show in Section 1, Figure 1 a qualitative plot of transportation of CelebA images. Because barycentric projection averages $\pi_{Y|X=x}$, output images are blurred and show visible mixing of multiple digits. By directly sampling the optimal transport plan, SCONES can separate these modes and generate more realistic images.

At low regularization levels, Algorithm 1 becomes more expensive and can become numerically unstable. As shown in Figures 3 and 1, SCONES can be used to sample the Sinkhorn coupling in intermediate regularization regimes, where optimal transport has a nontrivial effect despite $\pi_{Y|X=x}$ not concentrating on a single image.

|  | KL regularization, | | | $\chi^2$ regularization, | | |
|---|---|---|---|---|---|---|
|  | $\lambda = 0.1$ | $\lambda = 0.01$ | $\lambda = 0.005$ | $\lambda = 0.1$ | $\lambda = 0.01$ | $\lambda = 0.001$ |
| SCONES, Super-res. | 35.59 | 35.77 | 43.80 | 25.84 | 25.64 | 25.59 |
| Bary. Proj., Super-res. | 193.92 | 230.85 | 228.78 | 190.10 | 216.54 | 212.72 |
| SCONES, Identity | 36.62 | 34.84 | 43.99 | 25.51 | 25.65 | 27.88 |
| Bary. Proj., Identity | 195.64 | 217.24 | 217.67 | 188.29 | 219.96 | 214.90 |

Table 1: FID metric of samples generated by barycentric projection and SCONES, computed on $n = 5000$ samples from each model. For comparison to unregularized OT methods, we also trained a Wasserstein-2 GAN ($W_2$ GAN) [14] and a Wasserstein-2 Generative Network ($W_2$ Gen) [11]. $W_2$ GAN achieves FIDs 55.77 on the super-res. task and 32.617 on the identity task. $W_2$ Gen achieves FIDs 32.80 on the super-res. task and 20.57 on the identity task.

To quantitatively assess the quality of images generated by SCONES, we compute the FID scores of generated CelebA images on two optimal transport problems: transporting 2x downsampled CelebA images to CelebA (the 'Super-res.' task) and transporting CelebA to CelebA (the 'Identity' task) for a variety of regularization parameters. The FID score is a popular measurement of sample quality for image generative models and it is a proxy for agreement between the distribution of SCONES samples of the marginal $\pi_Y(y)$ and the true distribution $\tau(y)$. In both cases, we partition CelebA into two datasets of equal size and optimize Algorithm 1 using separated partitions as source and target data, resizing the source data in the super-resolution task. As shown in Table 1, SCONES has a significantly lower FID score than samples generated by barycentric projection. However, under ideal tuning, the unconditional score network generates CelebA samples with FID score 10.23 [24], so there is some cost in sample quality incurred when using SCONES.

## 5.2 Sampling Synthetic Data

To compare SCONES to a ground truth Sinkhorn coupling in a continuous setting, we consider entropy regularized optimal transport between Gaussian measures on $\mathbb{R}^d$. Given $\sigma = \mathcal{N}(\mu_1, \Sigma_1)$ and $\tau = \mathcal{N}(\mu_2, \Sigma_2)$, the Sinkhorn coupling of $\sigma$, $\tau$ is itself a Gaussian measure and it can be written in closed form in terms of the regularization $\lambda$ and the means and covariances of $\sigma$, $\tau$ [8]. In dimensions $d \in \{2, 16, 54, 128, 256\}$, we consider $\Sigma_1, \Sigma_2$ whose eigenvectors are uniform random (i.e. drawn from the Haar measure on $SO(d)$) and whose eigenvalues are sampled uniform i.i.d.

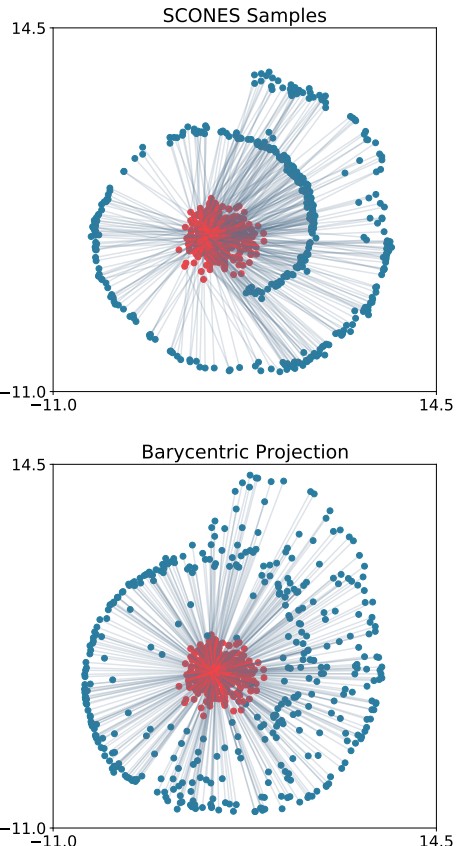

Figure 4: Entropy regularized, $\lambda = 2$, $L^2$ cost SCONES and BP samples on transportation from a unit Gaussian source distribution to the Swiss Roll target distribution. For many samples, the Barycentric average lies off the manifold of high target density, whereas SCONES can separate multiple modes of the conditional coupling and correctly recover the target distribution.

|        | $d = 2$ | $d = 16$ | $d = 64$ | $d = 128$ | $d = 256$ |
|--------|---------|----------|----------|-----------|-----------|
| SCONES | $0.025 \pm 0.0014$ | $0.52 \pm 0.0086$ | $1.2 \pm 0.014$ | $1.4 \pm 0.066$ | $2.0 \pm 0.047$ |
| BP     | $7.1 \pm 0.13$ | $35 \pm 0.23$ | $42 \pm 0.14$ | $41 \pm 0.088$ | $41 \pm 0.098$ |

Table 2: Comparison of SCONES to BP on KL-regularized optimal transport between random high-dimensional gaussians. In each cell, we report the average BW-UVP between a sample empirical covariance and the analytical solution. We report the average over $n = 10$ independent random source, target Gaussians and the standard error of the mean.

from $[1, 10]$. In all cases, we set means $\mu_1, \mu_2$ equal to zero and choose regularization $\lambda = 2d$. In the Gaussian setting, $\mathbb{E}[\|x - y\|_2^2]$ is of order $d$, so this choice of scaling ensures a fair comparison across problem dimensions by fixing the relative magnitudes of the cost and regularization terms.

We evaluate performance on this task using the Bures-Wasserstein Unexplained Variance Percentage [4], BW-UV$(\hat{\pi}, \pi^\lambda)$, where $\pi^\lambda$ is the closed form solution given by Janati et al. [8] and where $\hat{\pi}$ is the joint empirical covariance of $k = 10000$ samples $(x, y) \sim \pi$ generated using either SCONES or Barycentric Projection. We train SCONES according to Algorithm 1 and generate samples according to Algorithm 2. In place of a score estimate, we use the ground truth target score $\nabla_y \log \tau(y) = \Sigma_2^{-1}(y - \mu_2)$ and omit annealing. We compare SCONES samples to the true solution in the BW-UVP metric [4] which is measured on a scale from 0 to 100, lower is better. We report BW-UVP$(\hat{\pi}, \pi^\lambda)$ where $\hat{\pi}$ is a $2d$-by-$2d$ joint empirical covariance of SCONES samples $(x, y) \sim \hat{\pi}$ or of BP samples $(x, T_\theta(x))$, $x \sim \sigma$ and $\pi^\lambda$ is the closed-form covariance.

## 6   Discussion and Future Work

We introduce and analyze the SCONES method for learning and sampling large-scale optimal transport plans. Our method takes the form of a conditional sampling problem for which the conditional score decomposes naturally into a prior, unconditional score $\nabla_y \log \tau(y)$ and a "compatibility term" $\nabla_y \log M(V(x, y))$. This decomposition illustrates a key benefit of SCONES: one score network may be re-used to cheaply transport many source distributions to the same target. In contrast, learned forward-model-based transportation maps require an expensive training procedure for each distinct pair of source and target distribution. This benefit comes in exchange of increased computational cost of iterative sampling. For example, generating 1000 samples requires roughly 3 hours using one NVIDIA 2080 Ti GPU. The cost to sample score-based models may fall with future engineering advances, but iterative sampling intrinsically require multiple forward pass evaluations of the score estimator as opposed to a single evaluation of a learned transportation mapping.

There is much future work to be done. First, we study only simple fully connected ReLU networks as parametrizations of the dual variables. Interestingly, we observe that under $L^2$ transportation cost, parametrization by multi-layer convolutional networks perform equally or worse than their FCN counterparts when optimizing Algorithm 1. One explanation may be the *permutation invariance* of $L^2$ cost: applying a permutation of coordinates to the source and target distribution does not change the optimal objective value and the optimal coupling is simply conjugated by a coordinate permutation. As a consequence, the optimal coupling may depend non-locally on input data coordinates, violating the inductive biases of localized convolutional filters. Understanding which network parametrizations or inductive biases are best for a particular choice of transportation cost, source distribution, and target distribution, is one direction for future investigation.

Second, it remains to explore whether there is a potential synergistic effect between Langevin sampling and optimal transport. Heuristically, as $\lambda \to 0$ the conditional plan $\pi_{Y|X=x}$ concentrates around the transport image of $x$, which should improve the mixing time required by Langevin dynamics to explore high density regions of space. In Section B of the Appendix, we prove a known result, that the entropy regularized $L^2$ cost compatibility term $M(V(x, y)) = e^{V(x,y)/\lambda}$ is a log-concave function of $y$ for fixed $x$. It the target distribution is itself log-concave, the conditional coupling $\pi_{Y|X=x}$ is also log-concanve and hence Langevin sampling enjoys exponentially fast mixing time. However, more work is required to understand the impacts of non-log-concavity of the target and of optimization errors when learning the compatibility and score functions in practice. We look forward to future developments on these and other aspects of large-scale regularized optimal transport.

## Acknowledgments and Disclosure of Funding

M.D. acknowledges funding from Northeastern University's Undergraduate Research & Fellowships office and the Goldwater Award. P.H. was supported in part by NSF awards 2053448, 2022205, and 1848087.

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
