# A    Regularizing Optimal Transport with $f$-Divergences

| Name | $f(v)$ | $f^*(v)$ | $f^{*\prime}$ | $\text{Dom}(f^*(v))$ |
|---|---|---|---|---|
| Kullback-Leibler | $v \log(v)$ | $\exp(v-1)$ | $\exp(v-1)$ | $v \in \mathbb{R}$ |
| Reverse KL | $-\log(v)$ | $\log(-\frac{1}{v}) - 1$ | $-\frac{1}{v}$ | $v < 0$ |
| Pearson $\chi^2$ | $(v-1)^2$ | $\frac{v^2}{4} + v$ | $\frac{v}{2} + 1$ | $v \in \mathbb{R}$ |
| Squared Hellinger | $(\sqrt{v} - 1)^2$ | $\frac{v}{1-v}$ | $(1-v)^{-2}$ | $v < 1$ |
| Jensen-Shannon | $-(v+1)\log(\frac{1+v}{2}) + v \log v$ | $\frac{e^x}{2-e^x}$ | $\frac{2x}{e^x-2} + x - \log(2 - e^x)$ | $v < \log(2)$ |
| GAN | $v \log(v) - (v+1)\log(v+1)$ | $-v - \log(e^{-v} - 1)$ | $(e^{-y} - 1)^{-1}$ | $v < 0$ |

Table 3: A list of $f$-Divergences, their Fenchel-Legendre conjugates, and the derivative of their conjugates. These functions determine the corresponding dual regularizers $H_f^*(v)$ and compatibility functions $M_f(v)$. We take definitions of each divergence from [21]. Note that there are many equivalent formulations as each $f(v)$ is defined only up to additive $c(t-1)$, $c \in \mathbb{R}$, and the resulting optimization problems are defined only up to shifting and scaling the objective.

Here are some general properties of $f$-Divergences which are also used in Section B. We provide examples of $f$-Divergences in Table 3. The specific forms of $H_f^*(v)$ and $M_f(v)$ are determined by $f(v)$, $f^*(v)$, and $f^{*\prime}(v)$, which can in turn be used to formulate Algorithms 1 and 2 for each divergence.

**Definition A.1** ($f$-Divergences). Let $f : \mathbb{R} \to \mathbb{R}$ be convex with $f(1) = 0$ and let $p, q$ be probability measures such that $p$ is absolutely continuous with respect to $q$. The corresponding $f$-Divergence is defined $D_f(p||q) = \mathbb{E}_q[f(\frac{dp(x)}{dq(x)})]$ where $\frac{dp(x)}{dq(x)}$ is the Radon-Nikodym derivative of $p$ w.r.t. $q$.

**Proposition A.2** (Strong Convexity of $D_f$). *Let $\mathcal{X}$ be a countable compact metric space. Fix $q \in \mathcal{M}_+(\mathcal{X})$ and let $\mathcal{P}_q(\mathcal{X})$ be the set of probability measures on $\mathcal{X}$ that are absolutely continuous with respect to $q$ and which have bounded density over $\mathcal{X}$. Let $f : \mathbb{R} \to \mathbb{R}$ be $\alpha$-strongly convex with corresponding $f$-Divergence $D_f(p||q)$. Then, the function $H_f(p) \coloneqq D_f(p||q)$ defined over $p \in \mathcal{P}_q(\mathcal{X})$ is $\alpha$-strongly convex in 1-norm: for $p_0, p_1 \in \mathcal{P}_q(\mathcal{X})$,*

$$H_f(p_1) \geq H_f(p_0) + \langle \nabla_p H_f(p_0), p_1 - p_0 \rangle + \frac{\alpha}{2}|p_1 - p_0|_1^2. \tag{2}$$

*Proof.* Define the measure $p_t = tp_1 + (1-t)p_0$. Then $H_f$ satisfies the following convexity inequality (Melbourne [20], Proposition 2).

$$H_f(p_t) \leq tH_f(p_1) + (1-t)H_f(p_0) - \alpha \left( t|p_1 - p_t|_{\text{TV}}^2 + (1-t)|p_0 - p_t|_{\text{TV}}^2 \right)$$

By assumption that $\mathcal{X}$ is countable, $|p - q|_{\text{TV}} = \frac{1}{2}|p - q|_1$. It follows that,

$$H_f(p_1) \geq H_f(p_0) + \frac{H_f(p_0 + t(p_1 - p_0)) - H_f(p_0)}{t} + \frac{\alpha}{2} \left( |p_1 - p_t|_1^2 + (t^{-1} - 1)|p_0 - p_t|_1^2 \right)$$

$$\geq H_f(p_0) + \frac{H_f(p_0 + t(p_1 - p_0)) - H_f(p_0)}{t} + \frac{\alpha}{2}|p_1 - p_t|_1^2$$

and, taking the limit $t \to 0$, the inequality (2) follows. $\qquad\square$

For the purposes of solving empirical regularized optimal transport, the technical conditions of Proposition A.2 hold. Additionally, note that $\alpha$-strong convexity of $f$ is sufficient but not necessary for strong convexity of $H_f$. For example, entropy regularization uses $f_{\text{KL}}(v) = v \log(v)$ which is not strongly convex over its domain, $\mathbb{R}_+$, but which yields a regularizer $H_{\text{KL}}(p) = \text{KL}(p||q)$ that is 1-strongly convex in $l_1$ norm when $q$ is uniform. This follows from Pinksker's inequality as shown in [22]. Also, if $f$ is $\alpha$-strongly convex over a subinterval $[a, b]$ of its domain, then Proposition A.2 holds under the additional assumption that $a \leq \frac{dp(x)}{dq(x)}(x) \leq b$ uniformly over $x \in \mathcal{X}$.

# B Proofs

For convenience, we repeat the main assumptions and statements of theorems alongside their proofs. First, we prove the following properties about $f$-divergences.

**Proposition**, 2.4 – Regularization with $f$-Divergences. *Consider the empirical setting of Definition 2.1. Let $f(v) : \mathbb{R} \to \mathbb{R}$ be a differentiable $\alpha$-strongly convex function with convex conjugate $f^*(v)$. Set $f^{*\prime}(v) = \partial_v f^*(v)$. Define the violation function $V(x, y; \varphi, \psi) = \varphi(x) + \psi(y) - c(x, y)$. Then,*

1. *The $D_f$ regularized primal problem $K_\lambda(\pi)$ is $\lambda\alpha$-strongly convex in $l_1$ norm. With respect to dual variables $\varphi \in \mathbb{R}^{|\mathcal{X}|}$ and $\psi \in \mathbb{R}^{|\mathcal{Y}|}$, the dual problem $J_\lambda(\varphi, \psi)$ is concave, unconstrained, and $\frac{1}{\lambda\alpha}$-strongly smooth in $l_\infty$ norm. Strong duality holds: $K_\lambda(\pi) \geq J_\lambda(\varphi, \psi)$ for all $\pi$, $\varphi$, $\psi$, with equality for some triple $\pi^*, \varphi^*, \psi^*$.*

2. *$J_\lambda(\varphi, \psi)$ takes the form*

$$J_\lambda(\varphi, \psi) = \mathbb{E}_\mu[\varphi(x)] + \mathbb{E}_\sigma[\psi(y)] - \mathbb{E}_{\mu \times \sigma}[H_f^*(V(x, y; \varphi, \psi))]$$

*where $H_f^*(v) = \lambda f^*(\lambda^{-1} v)$.*

3. *The optimal solutions $(\pi^*, \varphi^*, \psi^*)$ satisfy*

$$\pi^*(x, y) = M_f(V(x, y; \varphi, \psi))\mu(x)\sigma(y)$$

*where $M_f(x, y) = f^{*\prime}(\lambda^{-1} v)$.*

*Proof.* By assumption that $f$ is differentiable, $K_\lambda(\pi)$ is continuous and differentiable with respect to $\pi \in \mathcal{M}_+(\mathcal{X} \times \mathcal{Y})$. By Proposition A.2, it is $\lambda\alpha$-strongly convex in $l_1$ norm. By the Fenchel-Moreau theorem, $K_\lambda(\pi)$ therefore has a unique minimizer $\pi^*$ satisfying strong duality, and by [10, Theorem 6], the dual problem is $\frac{1}{\lambda\alpha}$-strongly smooth in $l_\infty$ norm.

The primal and dual are related by the Lagrangian $\mathcal{L}(\pi, \varphi, \psi)$,

$$\mathcal{L}(\varphi, \psi, \pi) = \mathbb{E}_\pi[c(x, y)] + \lambda H_f(\pi) + \mathbb{E}_\mu[\varphi(x)] - \mathbb{E}_\pi[\varphi(x)] + \mathbb{E}_\sigma[\varphi(y)] - \mathbb{E}_\pi[\psi(y)] \quad (3)$$

which has $K_\lambda(\pi) = \max_{\varphi, \psi} \mathcal{L}(\varphi, \psi, \pi)$ and $J_\lambda(\varphi, \psi) = \min_\pi \mathcal{L}(\varphi, \psi, \pi)$. In the empirical setting, $\pi$, $\mu$, $\sigma$ may be written as finite dimensional vectors with coordinates $\pi_{x,y}$, $\mu_x$, $\sigma_y$ for $x, y \in \mathcal{X} \times \mathcal{Y}$. Minimizing the $\pi$ terms of $J_\lambda$,

$$\min_{\pi \in \mathcal{M}(\mathcal{X} \times \mathcal{Y})} \left\{ \mathbb{E}_\pi[c(x, y) - \varphi(x) - \psi(y)] + \lambda \mathbb{E}_{\mu \times \sigma}\left[ f\left( \frac{d\pi(x, y)}{d\mu(x)d\sigma(y)} \right) \right] \right\}$$

$$= \sum_{x,y \in \mathcal{X} \times \mathcal{Y}} - \max_{\pi_{x,y} \geq 0} \left\{ \pi_{x,y} \cdot (\varphi(x) + \psi(y) - c(x, y)) - \lambda\mu_x\sigma_y f\left( \frac{\pi_{x,y}}{\mu_x\sigma_y} \right) \right\}$$

$$= \sum_{x,y \in \mathcal{X} \times \mathcal{Y}} -h_{x,y}^*(\varphi(x) + \psi(y) - c(x, y))$$

where $h_{x,y}^*$ is the convex conjugate of $(\lambda\mu_x\sigma_y) \cdot f(p/(\mu_x\sigma_y))$ w.r.t. the argument $p$. For general convex $f(p)$, it is true that $[\lambda f(p)]^*(v) = \lambda f^*(\lambda^{-1} v)$ [3, Chapter 3]. Applying twice,

$$[(\lambda\mu_x\sigma_y) \cdot f(p/(\mu_x\sigma_y))]^*(v) = \lambda[(\mu_x\sigma_y)f(p/(\mu_x\sigma_y))]^*(\lambda^{-1} v) = (\lambda\mu_x\sigma_y) \cdot f^*(v/\lambda)$$

so that

$$\min_{\pi \in \mathcal{M}_+(\mathcal{X} \times \mathcal{Y})} \mathbb{E}_\pi[c(x, y) - \varphi(x) - \psi(y)] + \lambda \mathbb{E}_{\mu \times \sigma}\left[ f\left( \frac{d\pi(x, y)}{d\mu(x)d\sigma(y)} \right) \right]$$

$$= \sum_{x,y \in \mathcal{X} \times \mathcal{Y}} \mu_x\sigma_y \lambda f^*(\lambda^{-1} v)$$

$$= - \mathbb{E}_{\mu \times \sigma}[H_f^*(V(x, y; \varphi, \psi))]$$

for $H_f^*(v) = \lambda f^*(\lambda^{-1} v)$. The claimed form of $J_\lambda(\varphi, \psi)$ follows.

Additionally, for general convex $f(p)$, it is true that $\partial_v f^*(v) = \arg\max_p \{\langle v, p \rangle - f(p)\}$, [3, Chapter 3]. For $\varphi^*, \psi^*$ maximizing $J_\lambda(\varphi, \psi)$, it follows by strong duality that

$$
\begin{aligned}
\pi^*_{x,y} &= \underset{\pi \in \mathcal{M}_+(\mathcal{X} \times \mathcal{Y})}{\arg\min} \mathcal{L}(\varphi^*, \psi^*, \pi) \\
&= \nabla_V \mathbb{E}_{\mu \times \sigma}[H_f^*(V(x, y; \varphi^*, \psi^*))] = M_f(V(x, y; \varphi^*, \psi^*))\mu_x \sigma_y.
\end{aligned}
$$

as claimed. $\qquad\square$

We proceed to proofs of the theorems stated in Section 4.

**Assumption**, 4.1 – Approximate Linearity. *Let $f_\theta(x)$ be a neural network and set $\mathcal{K}_\theta(x) = [J_\theta^f(x)][J_\theta^f(x)]^T$ where $J_\theta^f(x)$ is the Jacobian of $f_\theta(x)$ with respect to $\theta$. Let $\Theta$ be a set of feasible weights, for example those reachable by gradient descent. Then $f_\theta(x)$ must satisfy,*

1. *There exists $R \gg 0$ so that $\Theta \subseteq B(0, R)$, where $B(0, R)$ is the Euclidean ball of radius $R$.*

2. *There exist $\rho_M > \rho_m > 0$ such that for $\theta \in \Theta$ and for all data points $\{X_i\}_{i=1}^N$,*
$$
\rho_M \geq \lambda_{max}(\mathcal{K}_\theta(X_i)) \geq \lambda_{min}(\mathcal{K}_\theta(X_i)) \geq \rho_m > 0.
$$

3. *For $\theta \in \Theta$ and for all data points $\{X_i\}_{i=1}^N$, the Hessian matrix $D_\theta^2 f_\theta(x_i)$ is bounded in spectral norm:*
$$
\|D_\theta^2 f_\theta(x_i)\| \leq \frac{\rho_M}{C_h}
$$
*where $C_h \gg 0$ depends only on $R$, $N$, and the regularization $\lambda$.*

The constant $C_h$ may depend on the dataset size $N$, the upper bound of $\rho_M$ for eigenvalues of the NTK, the regularization parameter $\lambda$, and it may also depend indirectly on the bound $R$.

**Theorem**, 4.2 – Optimizing Neural Nets. *Suppose $J_\lambda(\varphi, \psi)$ is $\frac{1}{s}$-strongly smooth in $l_\infty$ norm. Let $\varphi_\theta, \psi_\theta$ be neural networks satisfying Assumption 4.1 for the dataset $\{(x_i, y_i)\}_{i=1}^N$, $N = |\mathcal{X}| \cdot |\mathcal{Y}|$.*

*Then gradient descent of $J_\lambda(\varphi_\theta, \psi_\theta)$ with respect to $\theta$ at learning rate $\eta = \frac{\lambda}{2\rho_M}$ converges to an $\epsilon$-approximate global maximizer of $J_\lambda$ in at most $\left(\frac{2\kappa R^2}{s}\right)\epsilon^{-1}$ iterations, where $\kappa = \frac{\rho_M}{\rho_m}$.*

*Proof.* For indices $i$, let $S_{\theta_i} = (\varphi_{\theta_i}, \psi_{\theta_i})$ so that Assumption 4.1 applies with $S_\theta$ in place of $f_\theta$.

**Lemma B.1** (Smoothness). *$J_\lambda(S_\theta)$ is $\frac{2\rho_M}{s}$-strongly smooth in $l_2$ norm with respect to $\theta$:*
$$
J_\lambda(S_{\theta_2}) \leq J_\lambda(S_{\theta_1}) + \langle \nabla_\theta J_\lambda(S_{\theta_1}), S_{\theta_2} - S_{\theta_1} \rangle + \frac{\rho_M}{\lambda}\|\theta_2 - \theta_1\|_2^2.
$$

*Proof.* It is assumed that $J_\lambda(S)$ is $(\frac{1}{s}, l_\infty)$-strongly smooth and that $K_\lambda(\pi)$ is $(s, l_1)$-strongly convex. Note that $(\frac{1}{s}, l_2)$-strong smoothness is *weakest* in the sense that it is implied via norm equivalence by $(\frac{1}{s}, l_q)$-strong smoothness for $2 \leq q \leq \infty$.

$$
J_\lambda(S_2) \leq J_\lambda(S_1) + \langle \nabla_S J_\lambda(S_1), S_2 - S_1 \rangle + \frac{1}{2s} \mid S_2 - S_1 \mid_q^2
$$

$$
\implies J_\lambda(S_2) \leq J_\lambda(S_1) + \langle \nabla_S J_\lambda(S_1), S_2 - S_1 \rangle + \frac{1}{2s}\|S_2 - S_1\|_2^2
$$

A symmetric property holds for $(s, l_2)$-strong convexity of $K_\lambda(\pi)$ which is implied by $(s, l_p)$-strong convexity, $1 \leq p \leq 2$. By Assumption 4.1,

$$
J_\lambda(S_{\theta_2}) - J_\lambda(S_{\theta_1}) - \langle \nabla_S J_\lambda(S_{\theta_1}), S_{\theta_2} - S_{\theta_1} \rangle \leq \frac{1}{2s}\|S_{\theta_2} - S_{\theta_1}\|_2^2 \leq \frac{\rho_M}{2s}\|\theta_2 - \theta_1\|_2^2. \quad (4)
$$

To establish smoothness, it remains to bound $\langle \nabla_S J_\lambda(S_{\theta_1}), S_{\theta_2} - S_{\theta_1} \rangle$. Set $v = \nabla_S J_\lambda(S_{\theta_1}) \in \mathbb{R}^n$ and consider the first-order Taylor expansion in $\theta$ of $\langle v, S_\theta \rangle$ evaluated at $\theta = \theta_2$. Applying Lagrange's form of the remainder, there exists $0 < c < 1$ such that

$$
\begin{aligned}
\langle v, S_{\theta_2} \rangle = \langle v, S_{\theta_1} \rangle &+ \langle v, J_\theta^S(S_{\theta_2} - S_{\theta_1}) \rangle \\
&+ \frac{1}{2}\sum_{i=1}^n v_i(\theta_2 - \theta_1)^T[D_\theta^2(S_{\theta_1}(x_i) + c(S_{\theta_2}(x_i) - S_{\theta_1}(x_i)))](\theta_2 - \theta_1)
\end{aligned}
$$

and so by Cauchy-Schwartz,

$$\langle v, S_{\theta_2} - S_{\theta_1} \rangle \leq \langle v, J_\theta^S(S_{\theta_2} - S_{\theta_1}) \rangle + \frac{\|D_\theta^2\|}{2}\sqrt{N}\|v\|_2\|\theta_2 - \theta_1\|_2^2 \leq \frac{\rho_M}{2s}\|\theta_2 - \theta_1\|_2^2.$$

The final inequality follows by taking $C_h \geq \lambda\sqrt{N}\sup_v \|v\|_2$. This supremum is bounded by assumption that $\Theta \subseteq B(0, R)$. Plugging in $v = \nabla_S J_\lambda(S_{\theta_1})$, we have

$$\langle \nabla_S J_\lambda(S_{\theta_1}), S_{\theta_2} - S_{\theta_1} \rangle \leq \langle \nabla_S J_\lambda(S_{\theta_1}), J_\theta^S(S_{\theta_2} - S_{\theta_1}) \rangle + \frac{\rho_M}{2s}\|\theta_2 - \theta_1\|_2^2$$

$$= \langle \nabla_\theta J_\lambda(S_{\theta_1}), \theta_2 - \theta_1 \rangle + \frac{\rho_M}{2s}\|\theta_2 - \theta_1\|_2^2.$$

Returning to (4), we have

$$J_\lambda(S_{\theta_2}) - J_\lambda(S_{\theta_1}) \leq \langle \nabla_\theta J_\lambda(S_{\theta_1}), \theta_2 - \theta_1 \rangle + \frac{\rho_M}{s}\|\theta_2 - \theta_1\|_2^2.$$

from which Lemma B.1 follows. $\square$

**Lemma B.2** (Gradient Descent). *Gradient descent over the parameters $\theta$ with learning rate $\eta = \frac{s}{2\rho_M}$ converges in $T$ iterations to parameters $\theta_t$ satisfying $J_\lambda(S_{\theta_t}) - J_\lambda(S^*) \leq \left(\frac{2\kappa R^2}{s}\right)\frac{1}{T}$ where $\kappa = \frac{\rho_M}{\rho_m}$ is the condition number.*

*Proof.* Fix $\theta_0$ and set $\theta_{t+1} = \theta_t - \eta\nabla_\theta J_\lambda(S_\theta)$. The step size $\eta$ is chosen so that by Lemma B.1, $J_\lambda(S_t) - J_\lambda(S_{t+1}) \geq \frac{s}{2\rho_M}\|\nabla_\theta J_\lambda(S_{\theta_t})\|_2^2$.

By convexity, $J_\lambda(S^*) \geq J_\lambda(S_{\theta_t}) + \langle \nabla_S J_\lambda(S_{\theta_t}), S^* - S_{\theta_t} \rangle$, so that

$$\|\nabla_\theta J_\lambda(S_{\theta_t})\|_2^2 \geq \rho_m\|\nabla_S J_\lambda(S_{\theta_t})\|_2^2 \geq (J_\lambda(S_{\theta_t}) - J_\lambda(S^*))^2\left(\frac{\rho_m}{\|S_{\theta_t} - S^*\|_2^2}\right).$$

Setting $\Delta_t = J_\lambda(S_{\theta_t}) - J_\lambda(S^*)$, this implies $\Delta_t \geq \Delta_{t+1} + \Delta_t^2\left(\frac{s\rho_m}{2\rho_M\|S_{\theta_t} - S^*\|_2^2}\right)$ and thus $\Delta_t \leq \left[T\left(\frac{s\rho_m}{2\rho_M\|S_{\theta_t} - S^*\|_2^2}\right)\right]^{-1}$. The claim follows from $\|S_{\theta_t} - S^*\|_2 < R$. $\square$

Theorem 4.2 follows immediately from Lemmas B.1 and B.2. $\square$

**Theorem**, 4.3 – Stability of Regularized OT Problem. *Suppose $K_\lambda(\pi)$ is $s$-strongly convex in $l_1$ norm and let $\mathcal{L}(\varphi, \psi, \pi)$ be the Lagrangian of the regularized optimal transport problem. For $\hat{\varphi}, \hat{\psi}$ which are $\epsilon$-approximate maximizers of $J_\lambda(\varphi, \psi)$, the pseudo-plan $\hat{\pi} = M_f(V(x, y; \hat{\varphi}, \hat{\psi}))\mu(x)\sigma(y)$ satisfies*

$$|\hat{\pi} - \pi^*|_1 \leq \sqrt{\frac{2\epsilon}{s}} \leq \frac{1}{s}\left|\nabla_{\hat{\pi}}\mathcal{L}(\hat{\varphi}, \hat{\psi}, \hat{\pi})\right|_1.$$

*Proof.* For indices $i$, denote by $S_i$ the tuple $(\varphi_i, \psi_i, \pi_i)$. The regularized optimal transport problem has Lagrangian $\mathcal{L}(\varphi, \psi, \pi)$ given by

$$\mathcal{L}(\varphi, \psi, \pi) = \mathbb{E}_\pi[c(x, y)] + \lambda H_f(\pi) + \mathbb{E}_\mu[\varphi(x)] - \mathbb{E}_\pi[\varphi(x)] + \mathbb{E}_\sigma[\varphi(y)] - \mathbb{E}_\pi[\psi(y)]$$

Because $\mathcal{L}(\varphi, \psi, \pi)$ is a sum of $K_\lambda(\pi)$ and linear terms, the Lagrangian inherits $s$-strong convexity w.r.t. the argument $\pi$:

$$\mathcal{L}(S_2) \geq \mathcal{L}(S_1) + \langle \nabla\mathcal{L}(S_1), S_2 - S_1 \rangle + \frac{s}{2}|\pi_2 - \pi_1|_1^2.$$

Letting $S^* = (\varphi^*, \psi^*, \pi^*)$ be the optimal solution and $\hat{S} = (\hat{\varphi}, \hat{\psi}, \hat{\pi})$ be an $\epsilon$-approximation, it follows that

$$\epsilon \geq \mathcal{L}(\hat{S}) - \mathcal{L}(S^*) \geq \frac{s}{2}|\hat{\pi} - \pi^*|_1^2 \implies |\hat{\pi} - \pi^*| \leq \sqrt{\frac{2\epsilon}{s}}. \tag{5}$$

Additionally, note that strong convexity implies a Polyak-Łojasiewicz (PL) inequality w.r.t. $\hat{\pi}$.

$$s\left(\mathcal{L}(\hat{S}) - \mathcal{L}(S^*)\right) \leq \frac{1}{2}|\nabla_\pi\mathcal{L}(\hat{S})|_1^2. \tag{6}$$

The second inequality follows from (5) and the PL inequality (6).

$\square$

## B.1 Statistical Estimation of Sinkhorn Plans

We consider consider estimating an entropy regularized OT plan when $\mathcal{Y} = \mathcal{X}$. Let $\hat{\mu}$, $\hat{\sigma}$ be empirical distributions generated by drawing $n \geq 1$ i.i.d. samples from $\mu$, $\sigma$ respectively. Let $\pi_n^\lambda$ be the Sinkhorn plan between $\hat{\mu}$ and $\hat{\sigma}$ at regularization $\lambda$, and let $D := \mathrm{diam}(\mathcal{X})$. For simplicity, we also assume that $\mu$ and $\sigma$ are sub-Gaussian. We also assume that $n$ is fixed. Under these assumptions, we will show that $W_1(\pi_n^\lambda, \pi^\lambda) \lesssim n^{-1/2}$.

The following result follows from Proposition E.4 and E.5 of of Luise et al. [18] and will be useful in deriving the statistical error between $\pi_n^\lambda$ and $\pi^\lambda$. This result characterizes fast statistical convergence of the Sinkhorn potentials as long as the cost is sufficiently smooth.

**Proposition B.3.** *Suppose that $c \in \mathcal{C}^{s+1}(\mathcal{X} \times \mathcal{X})$. Then, for any $\mu, \sigma$ probability measures supported on $\mathcal{X}$, with probability at least $1 - \tau$,*

$$\|v - v_n\|_\infty, \|u - u_n\|_\infty \lesssim \frac{\lambda e^{3D/\lambda} \log 1/\tau}{\sqrt{n}},$$

*where $(u, v)$ are the Sinkhorn potentials for $\mu, \sigma$ and $(u_n, v_n)$ are the Sinkhorn potentials for $\hat{\mu}, \hat{\sigma}$.*

Let $\pi_n^\lambda = M_n \mu_n \sigma_n$ and $\pi^\lambda = M\mu\sigma$, We recall that

$$M(x, y) = \frac{1}{e} \exp\left(\frac{1}{\lambda}(\varphi(x) + \psi(y) - c(x, y))\right),$$

$$M_n(x, y) = \frac{1}{e} \exp\left(\frac{1}{\lambda}(\varphi_n(x) + \psi_n(y) - c(x, y))\right),$$

We note that $M$ and $M_n$ are uniformly bounded by $e^{3D/\lambda}$ [18] and $M$ inherits smoothness properties from $\varphi$, $\psi$, and $c$.

We can write (for some optimal, bounded, 1-Lipschitz $f_n$)

$$W_1(\pi_n^\lambda, \pi^\lambda) = |\int f_n \pi_n^\lambda - \int f_n \pi^\lambda|$$

$$\leq |\int f_n(H_n - H)\mu_n\sigma_n| + |\int f_n H(\mu_n\sigma_n - \mu\sigma)|$$

$$\leq |f_n|_\infty |H_n - H|_\infty + |\int f_n H(\mu_n\sigma_n - \mu\sigma)|. \tag{7}$$

If $\mu$ and $\sigma$ are $\beta^2$ subGaussian, then we can bound the second term with high probability:

$$\mathbb{P}\left(|\frac{1}{n^2}\sum_i\sum_j f_n(X_i, Y_j)H(X_i, Y_j) - \mathbb{E}_{\mu\times\sigma}f_n(X, Y)H(X, Y)| > t\right) < e^{-n^2\frac{t^2}{2\beta^2}}.$$

Setting $t = \sqrt{2}\log(\delta)\beta/n$ in this expression, we get that w.p. at least $1 - \delta$,

$$|\frac{1}{n^2}\sum_i\sum_j f_n(X_i, Y_j)H(X_i, Y_j) - \mathbb{E}_{\mu\times\sigma}f_n(X, Y)H(X, Y)| < \frac{\sqrt{2}\beta\log\delta}{n}.$$

Now to bound the first term in (7), we use the fact that $f_n$ is 1-Lipschitz and bounded by $D$. For the optimal potentials $\varphi$ and $\psi$ in the original Sinkhorn problem for $\mu$ and $\sigma$, we use the result of Proposition B.3 to yield

$$|H_n(x, y) - H(x, y)| = \left|\frac{1}{e}\exp\left(\frac{1}{\lambda}(\varphi_n(x) + \psi_n(y) - c(x, y))\right) - \frac{1}{e}\exp\left(\frac{1}{\lambda}(\varphi(x) + \psi(y) - c(x, y))\right)\right|$$

$$= \frac{1}{e}\left|\exp\left(\frac{1}{\lambda}(\varphi(x) + \psi(y) - c(x, y))\right)\left(1 - \exp\left(\frac{\varphi(x) - \varphi_n(x)}{\lambda}\right)\exp\left(\frac{\psi(y) - \psi_n(y)}{\lambda}\right)\right)\right|$$

$$\lesssim e^{3D/\lambda}|1 - e^{\frac{2}{\lambda\sqrt{n}}}|$$

$$\lesssim \frac{e^{3D/\lambda}}{\lambda\sqrt{n}}.$$

Thus, putting this all together,

$$W_1(\pi_n^\lambda, \pi^\lambda) \lesssim \frac{\mathsf{D}}{\sqrt{n}} + \frac{1}{n}.$$

Interestingly, the rate of estimation of the Sinkhorn plan breaks the curse of dimensionality. It must be noted, however, that the exponential dependence of Proposition B.3 on $\lambda^{-1}$ implies we can only attain these fast rates in appropriately large regularization regimes.

## B.2 Log-concavity of Sinkhorn Factor

The optimal entropy regularized Sinkhorn plan is given by

$$\pi^*(x, y) = \frac{1}{e} \exp\left(\frac{1}{\lambda}\left(\varphi^*(x) + \psi^*(y) - c(x, y)\right)\right) \mu(x)\sigma(y).$$

This implies that the conditional Sinkhorn density of $Y|X$ is

$$\pi^*(y|x) = \frac{1}{e} \exp\left(\frac{1}{\lambda}\left(\varphi^*(x) + \psi^*(y) - c(x, y)\right)\right) \sigma(y).$$

The optimal potentials satisfy fixed point equations. In particular,

$$\psi^*(y) = -\lambda \log \int \exp\left[-\frac{1}{\lambda}\left(c(x, y) - \varphi^*(x)\right)\right] d\mu(x).$$

Using this result, one can prove the following lemma.

**Lemma B.4** ([1]). *For the cost $\|x - y\|^2$, the map*

$$h(y) = \exp\left(\frac{1}{\lambda}\left(\varphi^*(x) + \psi^*(y) - \|x - y\|^2\right)\right)$$

*is log-concave.*

*Proof.* The proof comes by differentiating the map. We calculate the gradient,

$$\nabla \log h(y) = -2\frac{y - x}{\lambda} + \frac{2}{\lambda}\frac{\int \exp\left[-\frac{1}{\lambda}\left(\|x - y\|^2 - \varphi^*(x)\right)\right](y - x)d\mu(x)}{\int \exp\left[-\frac{1}{\lambda}\left(\|x - y\|^2 - \varphi^*(x)\right)\right]d\mu(x)}$$

and the Hessian,

$$\nabla^2 \log h(y) = -2\frac{I}{\lambda}$$
$$+ \frac{4}{\lambda^2}\frac{\int \exp\left[-\frac{1}{\lambda}\left(\|x - y\|^2 - \varphi^*(x)\right)\right](y - x)d\mu(x) \int \exp\left[-\frac{1}{\lambda}\left(\|x - y\|^2 - \varphi^*(x)\right)\right](y - x)^\top d\mu(x)}{(\int \exp\left[-\frac{1}{\lambda}\left(\|x - y\|^2 - \varphi^*(x)\right)\right]d\mu(x))^2}$$
$$- \frac{4}{\lambda^2}\frac{\int \exp\left[-\frac{1}{\lambda}\left(\|x - y\|^2 - \varphi^*(x)\right)\right](y - x)(y - x)^\top d\mu(x)}{\int \exp\left[-\frac{1}{\lambda}\left(\|x - y\|^2 - \varphi^*(x)\right)\right]d\mu(x)}$$
$$+ 2I/\lambda\frac{\int \exp\left[-\frac{1}{\lambda}\left(\|x - y\|^2 - \varphi^*(x)\right)\right]d\mu(x)}{\int \exp\left[-\frac{1}{\lambda}\left(\|x - y\|^2 - \varphi^*(x)\right)\right]d\mu(x)}$$
$$= -\frac{4}{\lambda^2}\left(-\frac{\int \exp\left[-\frac{1}{\lambda}\left(\|x - y\|^2 - \varphi^*(x)\right)\right](y - x)d\mu(x) \int \exp\left[-\frac{1}{\lambda}\left(\|x - y\|^2 - \varphi^*(x)\right)\right](y - x)^\top d\mu(x)}{(\int \exp\left[-\frac{1}{\lambda}\left(\|x - y\|^2 - \varphi^*(x)\right)\right]d\mu(x))^2}\right.$$
$$\left.+ \frac{\int \exp\left[-\frac{1}{\lambda}\left(\|x - y\|^2 - \varphi^*(x)\right)\right](y - x)(y - x)^\top d\mu(x)}{\int \exp\left[-\frac{1}{\lambda}\left(\|x - y\|^2 - \varphi^*(x)\right)\right]d\mu(x)}\right)$$

In the last term, we recognize that

$$\rho(x) = \frac{\exp\left[-\frac{1}{\lambda}\left(\|x - y\|^2 - \varphi^*(x)\right)\right]}{\int \exp\left[-\frac{1}{\lambda}\left(\|x - y\|^2 - \varphi^*(x)\right)\right]d\mu(x)}$$

forms a valid density with respect to $\mu$, and thus

$$\nabla^2 \log h(y) = -\frac{4}{\lambda^2}\mathsf{Cov}_{\rho d\mu}(X - y)$$

where we take the covariance matrix of $X - y$ with respect to the density $\rho d\mu$. $\qquad\square$

Suppose, for sake of argument, that $\sigma(y)$ is $\alpha$ strongly log-concave, and the function $h(y)$ is $\beta$ strongly log-concave. Then, $\pi_{Y|X=x} \propto h(y)\sigma(y)$, $\alpha + \beta$ strongly log-concave. In particular, standard results on the mixing time of the Langevin diffusion implies that the diffusion for $\pi_{Y|X=x}$ mixes faster than the diffusion for the marginal $\sigma$ alone. Also, as $\lambda \to 0$, the function $h(y)$ concentrates around $\varphi_{OT}(x) + \psi_{OT}(y) - \|x - y\|^2$, where $\varphi_{OT}$ and $\psi_{OT}$ are the optimal transport potentials. In particular, if there exists an optimal transport map between $\mu$ and $\sigma$, then $h(y)$ concentrates around the unregularized optimal transport image $y = T(x)$.

## C   Experimental Details

### C.1   Network Architectures

Our method integrates separate neural networks playing the roles of *unconditional score estimator*, *compatibility function*, and *barycentric projector*. In our experiments each of these networks uses one of two main architectures: a fully connected network with ReLU activations, and an image-to-image architecture introduced by Song and Ermon [24] that is inspired by architectures for image segmentation.

For the first network type, we write "ReLU FCN, Sigmoid output, $w_0 \to w_1 \to \ldots \to w_k \to w_{k+1}$," for integers $w_i \geq 1$, to indicate a $k$-hidden-layer fully connected network whose internal layers use ReLU activations and whose output layer uses sigmoid activation. The hidden layers have dimension $w_1, w_2, \ldots, w_k$ and the network has input and output with dimension $w_0, w_{k+1}$ respectively.

For the second network type, we replicate the architectures listed in Song and Ermon [24, Appendix B.1, Tables 2 and 3] and refer to them by name, for example "NCSN $32^2$ px" or "NCSNv2 $32^2$ px."

Our implementation of these experiments may be found in the supplementary code submission.

### C.2   Image Sampling Parameter Sheets

**MNIST $\leftrightarrow$ USPS**: details for qualitative transportation experiments between MNIST and USPS in Figure 3 are given in Table 4.

**CelebA, Blur-CelebA $\to$ CelebA**: we sample $64^2$ px CelebA images. The Blur-CelebA dataset is composed of CelebA images which are first resized to $32^2$ px and then resized back to $64^2$ px, creating a blurred effect. The FID computations in Table **??** used a shared set of training parameters given in Table 5. The sampling parameters for each FID computation are given in Table 6.

**Synthetic Data**: details for the synthetic data experiment shown in Figure 2 are given in Table 7.

| Problem Aspect | Hyperparameters | Numbers and details |
|---|---|---|
| Source | Dataset | USPS [19] |
| | Preprocessing | None |
| Target | Dataset | MNIST [13] |
| | Preprocessing | Nearest neighbor resize to $16^2$ px. |
| Score Estimator | Architecture | NCSN $32^2$ px, applied as-is to $16^2$ px images. |
| | Loss | Denoising Score Matching |
| | Optimization | Adam, lr $= 10^{-4}$, $\beta_1 = 0.9$, $\beta_2 = 0.999$. No EMA of model parameters. |
| | Training | 40000 training iterations, 128 samples per minibatch. |
| Compatibility | Architecture | ReLU network with ReLU output activation, $256 \to 1024 \to 1024 \to 1$ |
| | Regularization | $\chi^2$ Regularization, $\lambda = 0.001$. |
| | Optimization | Adam, lr $= 10^{-6}$, $\beta_1 = 0.9$, $\beta_2 = 0.999$ |
| | Training | 5000 training iterations, 1000 samples per minibatch. |
| Barycentric Projection | Architecture | ReLU network with sigmoid output activation, $256 \to 1024 \to 1024 \to 256$. Input pixels are scaled to $[-1, 1]$ by $x \mapsto 2x - 1$. |
| | Optimization | Adam, lr $= 10^{-6}$, $\beta_1 = 0.9$, $\beta_2 = 0.999$ |
| | Training | 5000 training iterations, 1000 samples per minibatch. |
| Sampling | Annealing Schedule | 7 noise levels decaying geometrically, $\sigma_0 = 0.2154, \dots, \sigma_6 = 0.01$. |
| | Step size | $\epsilon = 5 \cdot 10^{-6}$ |
| | Steps per noise level | $T = 20$ |
| | Denoising? [9] | Yes |
| | $\chi^2$ SoftPlus threshold | $\alpha = 1000$ |

Table 4: Data and model details for the **USPS → MNIST** qualitative experiment shown in Figure 3. For **MNIST → USPS**, we use the same configuration with source and target datasets swapped.

| Problem Aspect | Hyperparameters | Numbers and details |
|---|---|---|
| Source | Dataset | CelebA or Blur-CelebA [17] |
| | Preprocessing | $140^2$ px center crop. 
 If Blur-CelebA: nearest neighbor resize to $32^2$ px. 
 Nearest neighbor resize to $64^2$ px. 
 Horizontal flip with probability 0.5. |
| Target | Dataset | CelebA [17] |
| | Preprocessing | $140^2$ px center crop. 
 Nearest neighbor resize to $64^2$ px. 
 Horizontal flip with probability 0.5. |
| Score Estimator | Architecture | NCSNv2 $64^2$ px. |
| | Loss | Denoising Score Matching |
| | Optimization | Adam, lr $= 10^{-4}$, $\beta_1 = 0.9$, $\beta_2 = 0.999$. 
 Parameter EMA at rate 0.999. |
| | Training | 210000 training iterations, 
 128 samples per minibatch. |
| Compatibility | Architecture | ReLU network with ReLU output activation, 
 $3 \cdot 64^2 \to 2048 \to \ldots \to 2048 \to 1$ (8 hidden layers). |
| | Regularization | Varies in $\chi^2$ reg., $\lambda \in \{0.1, 0.1, 0.001\}$, 
 and KL reg., $\lambda \in \{0.1, 0.01, 0.005\}$. |
| | Optimization | Adam, lr $= 10^{-6}$, $\beta_1 = 0.9$, $\beta_2 = 0.999$ |
| | Training | 5000 training iterations, 
 1000 samples per minibatch. |
| Barycentric Projection | Architecture | NCSNv2 $64^2$ px applied as-is for image generation. |
| | Optimization | Adam, lr $= 10^{-7}$, $\beta_1 = 0.9$, $\beta_2 = 0.999$ |
| | Training | 20000 training iterations, 
 64 samples per minibatch. |

Table 5: Training details for the **CelebA, Blur-CelebA $\to$ CelebA** FID experiment (Figure 2).

| Problem | Noise $(\sigma_1, \sigma_k)$ | Step Size | Steps | Denoising? [9] | $\chi^2$ SoftPlus Param. |
|---|---|---|---|---|---|
| $\chi^2$, $\lambda = 0.1$ | $(9, 0.01)$ | $15 \cdot 10^{-7}$ | $k = 500$ | Yes | $\alpha = 10$ |
| $\chi^2$, $\lambda = 0.01$ | ——"—— | | | | |
| $\chi^2$, $\lambda = 0.001$ | ——"—— | | | | |
| KL, $\lambda = 0.1$ | $(90, 0.1)$ | $15 \cdot 10^{-7}$ | $k = 500$ | Yes | – |
| KL, $\lambda = 0.01$ | ——"—— | | | | |
| KL, $\lambda = 0.005$ | $(90, 0.1)$ | $1 \cdot 10^{-7}$ | $k = 500$ | Yes | – |

Table 6: Sampling details for the **CelebA, Blur-CelebA $\to$ CelebA** FID experiment (Figure 2).

| Problem Aspect | Hyperparameters | Numbers and details |
|---|---|---|
| Source | Dataset | Gaussian in $\mathbb{R}^{784}$, Mean and covariance are that of MNIST |
| | Preprocessing | None |
| Target | Dataset | Unit gaussian in $\mathbb{R}^{784}$. |
| | Preprocessing | None |
| Score Estimator | Architecture | None (score is given by closed form) |
| Compatibility | Architecture | ReLU network with ReLU output activation, $784 \rightarrow 2048 \rightarrow 2048 \rightarrow 2048 \rightarrow 2048 \rightarrow 1$ |
| | Regularization | KL Regularization, $\lambda \in \{1, 0.5, 0.25\}$. |
| | Optimization | Adam, lr $= 10^{-6}$, $\beta_1 = 0.9$, $\beta_2 = 0.999$ |
| | Training | 5000 training iterations, 1000 samples per minibatch. |
| Sampling | Annealing Schedule | No annealing. |
| | Step size | $\epsilon = 5 \cdot 10^{-3}$ |
| | Mixing steps | $T = 1000$ |
| | Denoising? [9] | Not applicable. |

Table 7: Sampling and model details for the synthetic experiment shown in Figure 2.