# OpenReview forum: "Score-based Generative Neural Networks for Large-Scale Optimal Transport"
_NeurIPS.cc/2021/Conference — NeurIPS 2021 Poster_

### Official Review · Reviewer_zJ3u · 2021-07-06

**Rating:** 7
**Confidence:** 4

**Summary:**

The authors propose a method to sample from the regularized OT plan (Sinkhorn plan) via a score-based method and Langevin dynamics.

**Limitations And Societal Impact:**

Social impact is ok. About the limitations:

The proposed pipeline consists of several successive elements (OT plan, score function, sampling) which might be practically complex and hard to set up. This is a limitation and should be emphasized and discussed. Besides, it is unclear how each of these elements performs separately (it is not clearly discussed in the paper). In particular, the quantitative evidence that the procedure actually samples from the true Sinkhorn transport plan requires more discussion.

The computational complexity is large (4 days of computing on 16 NVIDIA p100 GPUs for CelebA experiment), but this is discussed fairly well.



**Main Review:**

**Originality**

The proposed method is original. Nevertheless, the main topic of this paper is OT but the only related paper discussed/compared to in this paper is Seguy et al. (2018). There are already a lot of other methods (primarily based on the unregularized OT), which outperform the BP method of Seguy et al. (2018), some even consider applications to images. I will list some of them in the next sections. Therefore, a comparison with some of them should probably be included. Most of them solve a slightly different problem (recover pure "unregularized" OT map), but it seems ok since the generative modeling applications the authors consider require roughly the same.

**Quality**

There are some questions, comments which probably should be discussed in detail.

(1) The authors use dense networks (potentials) which is a contrast to most works computing OT for images. Why do the authors not use Conv Nets?

(2.1) To train OT plan, the authors use entropy-regularized OT, the optimization of which is known to be practically unstable in the continuous case. For example, Seguy et al. (2018) in Section 5.3 considered exactly L2 regularization, not entropy. Most later works Leygonie et al. (2019), Li et al. (2020), Makkuva et al. (2020), Korotin et al. (2020) which either use regularized OT or compare their method to it, all primarily use L2 reg. as it is claimed to be more stable than entropy. Therefore, I have doubts about the practical stability of the continuous entropy-OT training procedure, especially in the dims which are considered by the authors (64x64x3). My concern is that exp(lambda^-1*(phi+psi-c)) term appearing in training explodes since, e.g., for lambda=0.01, we already have exp(100). Can the authors comment on this?

(2.2)  The main baseline Seguy et al. (2020) considered in this paper is rather outdated (a lot of newer related work exists), i.e. comparison just with it might seem unfair.  Comparison with barycentric projection (BP) baseline by Seguy et al. (2018) might seem unreasonable. BP uses conditional expectation E\pi(y|x) while this paper considers the entire conditional distribution \pi(y|x). One may say that it is obvious that the method outperforms BP. Besides, BP is also known to suffer from bias. In README.MD of the official repo of Seguy et al. (2018) there is an image demonstrating the performance of BP in 2D. Even in this toy example, it is seen how different the BP is from the true samples (RED vs. GREEN points).

(3) In the synthetic data experiment, it is not transparent how the ground truth optimal plan is computed. Besides, the values of Frechet distance themself are not interpretable. Could the authors please provide values of its normalized variant BW-UVP, see Fan et. al. (2020)? In general, probably a more thorough evaluation is needed on Gaussian synthetic data (e.g. in various dimensions 2,4,...), to understand the scalability of the method.

(4) How are the samples shown in Figures 1,2 chosen? Are they sequential iterations of Langevin dynamics? Please comment.

(5) As far as I understand, in the CelebA experiment, for each face its downsampled copy is available on the other dataset. Such pairing might simplify the problem. Probably, half of the CelebA dataset should be coupled with the downsampled other half to avoid any inductive bias in data.

(6) Experimental details are given in Appendix, some code is present in the supplementary. However, reproducing the results might be hard due to the high required computational power. It seems to me that the paper lacks "easy" experiments, e.g. those which a reader can briefly reproduce on a single GPU or even CPU. Why do I think this is essential? The authors clearly point out that, e.g., Celeba Experiment, requires 4 days on 8xP100 GPUs. Even not all research groups who might be interested in improving/using current work are capable of spending so much computational time on reproducing the experiments.

**Clarity**

The paper is well written in general. Some places, e.g., Figures 1, 2, could be enriched. It would be nice to see a toy qualitative performance on 2D examples for better transparency, e.g. how well do the samples match the true distribution? This seems to be a necessary attribute of all OT papers nowadays.

**Significance**

This is probably the first paper demonstrating the application of a continuous regularized OT plan at a truly large scale. The performance of the method is promising. Besides, the method is supported by the convergence/stability theoretical results.

I admit the contribution of the authors, but I think a lot of details should be clarified and a more thorough evaluation to be performed. I set the score based on the current state of the paper but will be happy to reconsider it after the rebuttal phase.

**Additional feedback**

Please do not use \mu and \sigma simultaneously to denote probability distributions. This is misleading for obvious reasons. (!)

**Post rebuttal**
I raise the score to 5.

**Post rebuttal v2**
I raise the score to 7, see my last post.


**Time Spent Reviewing:**

10 hours

---

> ### Author Response · Authors · 2021-08-11
> **R4 Special Response**
>
> Thank you for these comments. In response to some specific concerns,
>
> 1. We use dense potential functions for fair comparison to Seguy et al and because we observed broadly that CNNs perform on-par or worse than FCNNs when optimizing Algorithm 1. We suspect that CNNs would outperform FCNNs *if* the transport cost itself depends on convolutional features, for example a cost computed in the activation space of a convolutional network. This would be an interesting direction for future work.
>
> 2. re: (2.1), For image-to-image optimal transport, rather than $\mathcal{L}^2$ cost, we compute the *mean* squared error, ie $c(x, y) = (1/n)\|x-y\|^2$ where $x, y$ are $n$-dimensional. This is equivalent to a rescaling of the objective from $\mathbb{E}[\|x-y\|^2/n] + \lambda H(\pi)$ to rescaled $\mathbb{E}[\|x-y\|^2] + n \lambda H(\pi)$. Because $\lambda$ is implicitly rescaled to a larger value, we can stably train without an exploding exponential term. We will update the final version to explain this unambiguously. We believe this is the correct choice for replicating Seguy et al because we can closely replicate their results, whereas training with unscaled $\mathcal{L}^2$ cost is impossible due to the blowup of the exponential term.
>
> 3. re: (4), The samples in Figures 1, 2 are the final iterations of multiple independent runs of Langevin dynamics, not sequential iterations. We plan to add plots of sequential iterations of the sampling algorithm throughout the annealing process.
>
> 4. We address items (2.2, 3, 5, 6) in detail in our general response. To summarize: (2.2) We plan to add comparison to samples generated by other algorithms for unregularized OT, (3) we plan to add BW-UVP computation for transportation between synthetic Gaussians in varying dimensions, (5) we will recreate experiments with a source/target split of CelebA, and (6) despite the expense of exactly recreating Table 1, our method is typically much cheaper, and one could feasibly replicate basic experiments from scratch in 1 week with a desktop GPU. For these planned experiments, we ran preliminary versions which are reported in the general response.

---

> > ### Comment · Reviewer_zJ3u · 2021-08-19
> > **More clarification of the OT performance needed**
> >
> > I thank the authors for answering my questions. However, I still have a very serious concern regarding the performance of constituent elements of the model (OT method, score function, sampling via dynamics).
> >
> > According to the evaluation in the Gaussian case provided by the authors in the answer, the OT method provides very high values of BW-UVP error. For example, even in small dimension 16, the method fails to capture ~30% of the variance of the transport plan. Therefore, it remains unclear whether the OT method truly works, especially in high dimensions such as 64x64x3 (faces).
> >
> > To further explain my concern, consider the following counterexample. Assume that the OT method totally fails and recovers M(V(...)) as nearly constant. In this case, sampling y's via Langevin dynamics for each x will be the same and will completely depend only on the quality of the score function s(). If the score function is good, then the FID of generated samples will also be small.
> >
> > As the consequence, the FID metric used by the authors in the faces experiment might be small even if the model poorly recovers the OT plan. As quantitive evaluation in the authors' answer demonstrates, this indeed might be a problem: not the true plan is recovered due to large BW-UVP. Therefore, this question should be more carefully studied, especially because of the paper is about the application of the score-based method for OT.
> >
> > To even further strengthen my concern, I point to the comment of another reviewer who noted that the samples generated by OT & score are worse than those generated from noise. Taking my previous comments into account, it seems like the OT method even negatively affects the performance.
> >
> > Other: Can the authors please provide the BW-UVP value for the MNIST-Gaussian experiment (Figure 3).
> >
> > To conclude, I think this is a paper with nice theoretical ideas but very questionable practical performance, especially of the part related to OT.

---

> > > ### Author Response · Authors · 2021-08-28
> > > **Further clarification of OT performance**
> > >
> > > We thank you for your comment, which helped us to locate a conceptual problem in the formulation of BW-UVP which we used to assess our method. Please see our general response detailing SCONES's performance in a BW-UVP experiment for both random gaussians and the MNIST experiments.
> > >
> > > In regard to the FID of SCONES compared to unconditional CelebA sampling: heuristically, sampling the marginal of an OT coupling is more difficult than sampling an unconditional data distribution, so SCONES's performance is not necessarily surprising. This is consistent with the pattern than FID decreases slightly as regularization decreases. On this more challenging task, SCONES with $L^2$ regularization is significantly improving both Korotin et al [2] and Leygonie et al [3].

---

> > > > ### Comment · Reviewer_zJ3u · 2021-08-28
> > > > **Misunderstanding of my concern**
> > > >
> > > > Thanks for the additional experiments, but I think that there is a misunderstanding.
> > > >
> > > > In the answer, the authors consider transport between Gaussians $\mu_1$ and $\mu_2$ on N-dimensional space. If I correctly understand, the authors use BW-UVP metric the same way as in previous works for unregularized OT, i.e. estimate BW-UVP$(\hat{\mu_2}, \mu_2)$. This is assessing the quality of the computed **marginal** distribution $\hat{\mu_2}\approx\mu_2 .$
> > > >
> > > > What I suggested in my response is to use BW-UVP to assess the computed **entropy-regularized transport plan** (Sinkhorn coupling) $\hat{\pi}$, i.e. a distribution on 2*N-dimensional space approximating the true regularized plan $\pi^{\epsilon}$, i.e. solution of (1). More precisely, I meant to evaluate BW-UVP$(\hat{\pi},\pi^{\epsilon})$. This is important, because this entire regularized plan (not just the N-dimensional marginal!) is one of the constituent elements in the final proposed model. The ground truth Sinkhorn coupling for Gaussians is known explicitly, e.g., see Janati et al. (2020) "Entropic Optimal Transport between Unbalanced Gaussian Measures has a Closed-Form". The paper "Entropy-regularized 2-Wasserstein distance between Gaussian measures" by Mallasto et al. (2020) also seems relevant.
> > > >
> > > > I strongly encourage the authors to include such an evaluation and provide BW-UVP values (for Gaussians and Gaussian-MNIST experiment) during the remaining period. No baselines are needed: there are no other methods for computing Entropy's regularized plan in the continuous case anyway.
> > > >
> > > > The current experiment does not fully test the quality of the recovered regularized plan. Yet I still recommend including it in the paper. I slightly raise my score for this paper.

---

> > > > > ### Author Response · Authors · 2021-08-28
> > > > > **We are already computing BW-UVP of the joint coupling**
> > > > >
> > > > > The updated BW-UVP results in our latest general response are already computed as $\text{BW-UVP}(\hat{\pi},\pi^\epsilon)$ where $\hat{\pi}$ is a $2d$-by-$2d$ empirical covariance estimate from SCONES samples and $\pi^\epsilon$ as given by Janati et al. (2020). We apologize for any confusing references to unregularized optimal transport which could have made this aspect of the experiment unclear -- we edited the general response to include this point.

---

> > > > > > ### Comment · Reviewer_zJ3u · 2021-08-29
> > > > > > **Additional Images**
> > > > > >
> > > > > > Got it, thanks.
> > > > > >
> > > > > > Could the authors please provide additional visualizations and samples for CelebA experiment that they are going to include in the final draft? It would be nice to see them before setting the final score.

---

> > > > > > > ### Author Response · Authors · 2021-09-01
> > > > > > > **We added additional images in a general response**
> > > > > > >
> > > > > > > We have added the requested figures in our latest general response. Thank you for your thoughts and extended discussion of this work!

---

> > > > > > > > ### Comment · Reviewer_zJ3u · 2021-09-02
> > > > > > > > **Final feedback**
> > > > > > > >
> > > > > > > > The authors have explicitly answered my concerns and I again increase my score to 7. I think the paper is good enough for NeurIPS, assuming that additional clarifications, references, experiments, answers to reviewers are added to the final version.
> > > > > > > >
> > > > > > > > This is probably the first paper demonstrating convincing qualitative/quantitative practical performance of entropy-regularized optimal transport **plan** based on neural networks. To the best of my knowledge, previous papers mainly considered **cost** (Sanjabi et al.) or **map** (Seguy et al.) only. Besides, the method is supported by relevant and meaningful theoretical results. They seem to be correct but I did not rigorously check them.
> > > > > > > >
> > > > > > > > *I will be happy if this paper is accepted. However, I think the paper might still benefit from another (though **minor**) round of revision. Therefore, I won't be unhappy if this paper is rejected.*

---

### Official Review · Reviewer_jbX8 · 2021-07-15

**Rating:** 6
**Confidence:** 4

**Summary:**

Using score-based generative modeling and Langevin dynamics, this paper proposes a new method to sample from the Sinkhorn coupling distribution. The method first solves the dual problem of regularized optimal transport by parameterizing the dual variables with neural networks. After learning the dual variables, the coupling distribution is known up to a normalizing constant, and can therefore be sampled from using Langevin dynamics and a score-based model as the prior. Experiments demonstrate advantages over barycenter methods.

**Limitations And Societal Impact:**

No obvious limitations or negative societal impact to me.

**Main Review:**

Solving large scale optimal transport has important practical applications, and is of high relevance to the NeurIPS community. This paper proposes a new method to sample from the optimal coupling distribution.

Advantages of this submission:
1. Very clear writing.
2. I am not an expert on optimal transport, but the following theoretical contributions seem to be new: (i) generalizing regularized optimal transport to f-divergence regularizers; (ii) convergence rate of learning neural network based $\phi_\theta$ and $\psi_\theta$ models with gradient descent (under some rather strong assumptions).
3. The proposed SCONES method is very general, and can be applied to regularized optimal transport problems with any f-divergence regularizer. When a pre-trained score-model is available, SCONES is also quite efficient.

Disadvantages of the submission:
1. It is unclear whether samples from Langevin dynamics really capture the optimal coupling distribution in practice. Authors provide some sanity check on synthetic Gaussian distributions, but it is important to check on image problems as well. It is not hard anyways: why not just estimate the transportation cost using samples from SCONES, and compare it with the value of the dual problems obtained after learning $\phi_\theta$ and $\psi_\theta$ models. Given that authors only used 8 pages out of a 9-page limit, there should be enough space to include such an experiment.

2. Authors need to demonstrate more practical applications of sampling from the optimal coupling distribution. Why would we want to convert MNIST to USPS images and vice versa? Why should we use optimal transport to upsample images? It's not clear to me that optimal transport will provide correct upsampled images and can have any practical advantage over existing methods for superresolution.

Some minor issues:
1. "by providing" repeated twice on line 184.
2. Some broken references in the appendix.

Post rebuttal update:
Authors addressed some of my concerns in the response. I am happy to increase my score to 6.

**Time Spent Reviewing:**

2

---

> ### Author Response · Authors · 2021-08-11
> **R3 Special Response**
>
> Thanks for these comments -- please see the general response to all reviewers. We are planning to improve our sanity checks on Gaussian data to check that our method is really learning the optimal coupling in practice. We will also run the suggested experiment by computing the duality gap between objective value of Algorithm 1 and the average transport cost of samples from Algorithm 2.
>
> In regards to practical applications of SCONES: the usefulness of classical optimal transport algorithms in low dimension for applications like logistics or domain adaptation is indisputable, but there has been a limited set of tools for scaling these applications to high dimensions and large datasets. Our main practical goal for SCONES is to contribute one such tool, paving the road for future applications. From this perspective, we do not necessarily intend for SCONES to be a competitive algorithm for exactly solving upsampling, but rather we use upsampling as a test case to demonstrate that our method solves regularized OT between visually similar image datasets.

---

### Official Review · Reviewer_R3r7 · 2021-07-15

**Rating:** 6
**Confidence:** 3

**Summary:**

The authors propose methodology to sample from high dimensional regularized transport maps using Langevin diffusion with neural network parameterized score (via score-matching), and neural network parameterized potentials.

The authors derive novel quantitative convergence results regarding approximating dual potentials with wide neural nets and the subsequent quality of approximated transport map.


**Limitations And Societal Impact:**

They mention that FID is not as good as regular score matching generative modeling
There is limited societal impact compared to existing methods.


**Main Review:**

This paper uses NN parameterized dual potentials and NN parameterized score, both of which are not original, to sample from a transport map, which is the novel application. Arguably more interestingly, the authors derive quantitative convergence rates relating the NN parameterized dual potentials to quality of transport map approximation.

The experimental results are not very strong e.g. the upsampled CelebA results start from down sampled CelebA rather than noise and are of worse quality than using score-matching to sample images starting from noise.

It is not clear what the advantages of using score-matching + NN parameterized potentials are over just score-matching for generative models and conditional generation. However, the theoretical results and proof using NN parameterized potentials is very interesting and may be used in other OT related work.

Strengths:
Leverages existing pre-trained score networks for sampling high dimensional OT transport maps e.g. for conditional generation.
Theoretical contributions regarding using NN parameterized potentials for OT gives legitimacy to this approach which appears novel and significant

Weaknesses:
SCONES requires pre-trained score network, yet the method’s performance (for generative modeling) appears to be worse than simply using the pre-trained score network in terms of FID score.
Similar to previous score based methods, it requires large neural networks, and many diffusion steps resulting in slow sampling.

Missing references: Parameterizing potential with Neural networks, ICNN [1] or RKHS [2], flows [3]

The statement of “breaking curse of dimensionality” is tangential to the main work and is only true under strong sub-Gaussian assumption, line 544 in appendices.

The method appears expensive e.g. CelebA upsampling uses 16 GPUs for 4 days. Is this just for training the potential networks or both potential and score networks?

No comparison with other transport samplers rather than barycentric average is provided. See e.g. [3], I imagine there are others sampling the transport map rather than averaging.

Minor:
Notation:
$\lambda_\max$ not formally defined, I assume eigenvalues, line 193
Do you mean M or H in proof, appendices line 533-542
Typo using $S_{\theta_1}- S_{\theta_2}$ in line 485 of appendices but I assume means $\theta_1-\theta_2$ as in proof line 497

Careful with splitting main/supplementary,  I assume these will be corrected when files are combined:
Latex errors, links/refs e.g.  “??” missing in Appendices in main file
Link [15] in the supplementary should be [14], line 527
Comment with initials “MD” left in appendices on main file, line 425

[1] Optimal transport mapping via input convex neural networks,
[2] Stochastic Optimization for Large-scale Optimal Transport,
[3] Learning normalizing flows from Entropy-Kantorovich potentials,


**Time Spent Reviewing:**

15

---

> ### Author Response · Authors · 2021-08-11
> **R2 Special Response**
>
> Thanks for these comments. We will fix the recommended typos and add missing citations. Here are some responses to specific concerns:
>
> 1. re: Performance relative to the pre-trained network. The key comparison of our paper is between BP and SCONES, where we show that score based sampling helps eliminate the error induced by BP and vastly improves the FID score. Based on preliminary results (see our general response), SCONES with L2 regularization is also outperforming unconditional methods like Korotin et al [2] and Leygonie et al [3]. Heuristically, sampling the marginal of an OT coupling is more difficult than sampling an unconditional data distribution, so our diminished performance is not necessarily surprising. We hope that as the engineering maturity of SCONES reaches that of unconditional score-based sampling, it may be possible to close this performance gap.
>
> 2. re: Computational expense. Please see the general response -- '16 GPUs for 4 days' is technically correct, but it's an unrealistically high upper bound on the expense to replicate our method, which we will clarify in our final draft. Training and sampling all the networks for SCONES can be done from scratch in under a week on 1 desktop GPU.
>
> 3. re: Comparison to other samplers. We will add comparison to other samplers for unregularized optimal transport as described in the general response.

---

### Official Review · Reviewer_6aG5 · 2021-07-18

**Rating:** 6
**Confidence:** 4

**Summary:**

The authors consider a regularised optimal transport problem. They
- obtained a dual objective in case of a general regularisation based on f-divergence,
- used
a) the large-scale stochastic dual approach, introduced by Seguy et al., to solve the optimal transport problem in their generalised setting,
b)  Langevin Sampling to sample from the conditional distribution of Y given X,
- proved some convergence guarantees.

**Limitations And Societal Impact:**

The approach consists of several steps, and it is important to discuss limitations of different steps separately, it is unclear how these different steps perform separately.

**Main Review:**

- There are a number of recent papers with large-scale approaches to OT, e.g.
1) Jacob Leygonie, Jennifer She, Amjad Almahairi, Sai Rajeswar, and Aaron Courville. Adversarial
computation of optimal transport maps. arXiv preprint arXiv:1906.09691, 2019.
2) Alexander Korotin, Vage Egiazarian, Arip Asadulaev, Alexander Safin, and Evgeny Burnaev.
Wasserstein-2 generative networks. In International Conference on Learning Representations, 2021.
3) Ashok Vardhan Makkuva, Amirhossein Taghvaei, Sewoong Oh, and Jason D Lee. Optimal
transport mapping via input convex neural networks. arXiv preprint arXiv:1908.10962, 2019, etc.

These methods solve un-regularized OT and recover an OT map; they significantly outperform the approach of Seguy et al. (2018). Any comments? Any comparisons? Although the problem statement is slightly different, the comparison can be done using some generative modeling application. Moreover, it is easy to outperform the approach of Seguy et al. (2018), since it uses the barycentric projection (BP) baseline which is weak compared to using the entire conditional distribution pi(y|x).

- In general, the text of the paper is rather well written. However, some details are missing, see comments below.

- Can the authors prove their theoretical result about the convergence for CNNs? It seems not optimal to use FCNN for images, so CNNs are needed. The authors mentioned that the assumptions of the theorem are valid for FCNN. What is about CNN? Moreover, could the authors provide some more visual results of the experiment with Celeba? FCNN are not efficient for such high dimensionality (64x64x3), so it could be good to analyse failure cases.

- In case of experiments with synthetic Gaussian data I would like to see how the method scales with the dimension. Moreover, the authors provide the value of the Frechet distance. However, to understand whether they are big or not we need some characteristic value or normalisation. So the current representation of the results does not allow to comprehensively analyse them.

- The authors used the Langevin dynamics to generate images. Could the authors provide some intermediate results on how the dynamics converges?

- Protocols of experiments are not fully detailed, see comments above. Also, in case of the experiment with the Celeba dataset, how the samples X and Y were organised? For each image from X there is a high-res image of the same face in Y? If yes, this simplifies the problem. Also, I would prefer to see more  results of the corresponding super-resolution task, not only values of the FID score and some faces in Figure 1. Also, why not to include more toy examples and illustrate the convergence?

- The results on Celeba are promising, they demonstrates that we can construct OT rather efficiently even for rather high-res images.

- Overall, the results are promising and some theoretical guarantees are proved, but more details about the experiments are needed.  Also, comparisons with recent method are missing.

I need more details/comments to upgrade/downgrade the current rating.


**Time Spent Reviewing:**

3.0

---

> ### Author Response · Authors · 2021-08-11
> **R1 Special Response**
>
> Thank you for your comments. In response to some specific concerns,
> 1. We agree that it would be fair and informative to compare generative modeling of SCONES as in Table 1 to the performance of ICNN methods like Makkuva et al, Korotin et al, and generative modelling as in Leygonie et al. We will add this experiment, see our preliminary results in 'Changes to the final paper.'
>
> 2. We observed broadly that CNNs perform on-par or worse than FCNNs in Algorithm 1. We suspect that CNNs would outperform FCNNs *if* the transport cost itself depends on convolutional features, for example a cost computed in the activation space of a convolutional network. This is an interesting direction for future work.
>
> 3. re: Point 3, We will replace our synthetic Gaussian experiments with a computation of BW-UVP for regularized OT between two Gaussians in dimension 2, 16, 64, 128, 256. However, because of the fundamental differences between regularized and unregularized OT, we will not compare these synthetic results to the corresponding experiment for unregularized OT.
>
> 4. re: Point 4, we will add more figures illustrating the evolution of Langevin dynamics during sampling.
>
> 5. re: Point 5, we will recreate our experiments using disjoint CelebA images as source and target datasets. We expect this to have minimal impact on the results shown in Table 1.

---

> > ### Comment · Reviewer_6aG5 · 2021-09-02
> > **Decided to increase the score to 6 after analysing all discussions**
> >
> > - Unfortunately, due to other obligations I was not able to take part in the discussions. I really appreciate the hard work done by other reviewers in discussing the paper with the authors, especially the discussion of zJ3u with the authors
> >
> > - The authors clearly addressed my questions/comments
> >
> > - Now I see that the paper has not only some nice theoretical results, but the practical results of the paper are convincing thanks to new experiments. Thus I am convinced that the paper can be accepted and published in the proceedings

---

### Author Response · Authors · 2021-08-11
**General Response**

Thank you for closely reading our paper and providing this insightful feedback! In this section we will respond to some shared questions and comments.

### Questions about experiments
**Is the theory applicable to CNNs instead of FCNNs?**  Yes. We phrase Assumption 4.1 so that it holds generally for the broad class of network architecture satisfying an 'NTK limit.' This includes both CNNs and residual networks, see Allen-Zhu et al [1].

In comparison to related work based on Input Convex Neural Nets (ICNN), such as Makkuva et al [5] and Korotin et al [2], a strength of our approach is that we can prove finite size networks provide a sufficiently expressive function class to solve regularized OT (Theorem 4.2), whereas expressivity of the ICNN function class is typically assumed without justification. While Assumption 4.1 holds for a variety of *unconstrained* network architectures, it does not hold for ICNN, so our approach is not directly applicable to those settings.

**Why use FCNNs over CNNs experimentally?** We tried parametrizing the potential functions as CNNs and we observed equal or worse experimental performance compared to FCNN parametrization when optimizing Algorithm 1, for both the CelebA and MNIST-USPS experiments.

There is a good reason for this: with $\mathcal{L}^2$ ground cost, the optimal transport problem is permutation invariant, so it may depend non-locally on the coordinates of the input data. Applying a permutation of coordinates to the source and target distribution would not change the optimal objective value and the optimal coupling would simply be conjugated by a coordinate permutation. Conversely, there is no reason a-priori that the optimal potential functions should depend totally on localized convolutional features, and indeed we observed this is not the case. We will comment on this observation in our experimental details.

**On the computational expense of SCONES.**  In the final draft of our paper, we will clarify that training our method is quite cheap and sampling/evaluation is not inaccessibly expensive. As an example, using 1 2080 Ti desktop GPU, here is the wall clock times needed to recreate one instance of our CelebA superresolution task.

*  Optimizing density parameters (Algorithm 1): 34 minutes.
* Generating samples (Algorithm 2): 3 hours, 11 minutes per 1000 samples.
* Learning the score model: free when using pretrained. Otherwise, training from scratch requires a few days to a week, on the same order as the cost of training other generative models as in Leygonie et al [3], Makkuva et al [5].

Sampling the model is the computational bottleneck, but this time is rapidly diminishing with new engineering advances in score based modeling, and the method is already accessible for other groups to replicate in less than a week. The listed computation cost of '4 days on 16 GPUs' is only required to fully replicate the FID computations in Table 1 by generating 5,000 images for each of the 12 problem settings.

### Changes to the Final Draft
Based on the reviewers' shared concerns, we will make the following changes in the final draft.

**Use separated CelebA images for source and target data.** . We will ensure that the CelebA images used as source and target datasets are chosen from disjoint subsets of CelebA. We tested this change for entropy regularized CelebA superresolution (Table 1, row 1, column 1) and observed minimal change in FID: the new, old values are 34.604, 33.94 respectively. Based on this, we expect minimal changes to current Table 1 quantities in the final draft.

**Compare to unregularized OT.**  We will add two columns to Table 1 for comparison to algorithms proposed by Korotin et al [2] and Leygonie et al [3] for unregularized OT. Each column will show its method's FID on the CelebA superresolution and identity tasks. We ran a preliminary test of these methods on CelebA superresolution and observed FIDs 32.8, 55.77 respectively, both significantly worse than SCONES with L2 regularization (previously at-worst 25.75 FID with $\lambda=0.1$), while the first is on-par with entropy regularization.

**Report synthetic gaussian experiments in varying dimension with normalized metric.** We will replace our synthetic Gaussian experiment with a table of accuracies for entropy regularized OT between Gaussians in varying dimension. Rather than Frechet Distance, we will report its normalized variant, BW-UVP (Fan et al [4]).

We ran the following preliminary test. In dimension $d \in \{2, 16, 64, 128, 256\}$, we randomly generate source and target gaussians whose covariances have condition number at most $10$, then we use SCONES to sample the coupling with entropy regularization at $\lambda=1$. We observe BW-UVPs ~1.143, 28.87, 44.39, 46.76, and 48.12,~ where they range 0-100 and lower is better. (*Edit*: these BW-UVP scores are incorrect, please see our follow-up response)

**Add visualizations and samples.** We plan to add the following figures to the main body.

* Qualitative CelebA results showing the evolution of samples from noise. Specifically, we will show how conditional samples for an individual source image evolve from noise at different steps of the annealing process.

* Further qualitative CelebA results from Table 1. We will expand Figure 1 to add more visual results of the super-resolution and identity mapping tasks for CelebA.

*  Qualitative plot for a synthetic experiment. We will compare BP and SCONES for 2d qualitative datasets like 2moons, showing where source points are mapped in comparison to the target distribution.

Finally, we will include additional sample sheets in the appendix, which will show a large number of SCONES samples from CelebA for qualitative inspection.

### References
1. A Convergence Theory for Deep Learning via Over-Parameterization, Zeyuan Allen-Zhu et al.
2. Wasserstein-2 generative networks, Korotin et al.
3. Adversarial computation of optimal transport maps, Leygonie et al.
4. Scalable Computations of Wasserstein Barycenter via Input Convex Neural Networks, Fan et al.
5. Optimal transport mapping via input convex neural network, Makkuva et al.

---

### Author Response · Authors · 2021-08-28
**Our preliminary BW-UVP experiment did not accurately test our method, corrected experiment is favorable to SCONES.**

**Summary**: our previous BW-UVP incorrectly scales regularization with respect to dimension. In the corrected version, SCONES has BW-UVP values 0.0226, 0.574, 1.28, 1.26, 1.96 for $d = 2, 16, 64, 128, 256$ respectively. We also re-ran our original MNIST experiment, transporting MNIST's covariance to identity, and we observed BW-UVP of 2.14 for $d = 784$.

---

In our general response, we report preliminary BW-UVP for gaussian-to-gaussian transport under $L^2$ cost with entropy regularization. To be consistent prior work Korotin et al. Table 1, we held regularization fixed over all dimensions $d \in \{2, 16, 64, 128, 256\}$.

However, this choice is in contrast to our experimental setup, where we use a mean-squared-error cost $c(x, y) = (1/d) \|x-y\|^2$ where $x, y$ are $d$-dimensional. This is equivalent to rescaling the OT objective from $\mathbb{E}[|x-y|^2/d] + \lambda H(\pi)$ to rescaled $\mathbb{E}[|x-y|^2] + d \lambda H(\pi)$, and in particular, it is scaling the 'regularization level' with respect to problem dimension. By omitting this rescaling, our synthetic experiment was not accurately testing our method as it had been applied in other experiments.

We ran a corrected version with $L^2$ cost and regularization $d \lambda$ in place of $\lambda$. Using SCONES, we observed BW-UVP values 0.0226, 0.574, 1.28, 1.26, 1.96 for $d = 2, 16, 64, 128, 256$ respectively. We also re-ran our original MNIST experiment, transporting MNIST's covariance to identity, and we observed BW-UVP of 2.14 for $d = 784$. In contrast, the LSOT method achieves BW-UVP of 5.59, 37.5, 42.2, 41.1, 40.7, 91.8 for the $d = 2, 16, 64, 128, 256, 784$ problems respectively.

Therefore, in the synthetic setting that accurately represents our experiments, SCONES is both significantly improving BP and achieving similar BW-UVP scores as unregularized OT methods.

Morally speaking, we believe scaled regularization as $d \lambda$ is the correct choice in settings like our gaussian-to-gaussian where $\mathbb{E}[c(x, y)]$ is also scaling linearly in dimension, so that the two terms of the regularized objective are balanced and holding $\lambda$ fixed is a fair comparison between settings in different dimensions. Otherwise, increasing dimension is equivalent to decreasing regularization and our theory predicts numerous bad outcomes (eg. requiring larger networks and longer training times [Theorem 4.2], exponentially increased sample complexity [Supp Proposition B.3]) which account for the previously reported performance of SCONES.

Edit: to clarify an aspect of this experiment: we are reporting $\text{BW-UVP}(\hat{\pi},\pi^\epsilon)$ where $\hat{\pi}$ is a $2d$-by-$2d$ joint empirical covariance for SCONES samples and $\pi^\epsilon$ is given by [1]. To compute the corresponding score for BP, we compute the empirical covariance of samples $(x, T(x)) \in \mathbb{R}^{2d}$ where $x$ is drawn from the source distribution and $T(x)$ is the learned BP mapping.

[1] Entropic Optimal Transport between Unbalanced Gaussian Measures has a Closed Form, Janati et al. (2020)

---

### Author Response · Authors · 2021-09-01
**Additional visualizations and samples for CelebA experiment**

We prepared the additional figures for our CelebA experiments which we are planning to add to the body of our paper. You can find them here:

https://anonymous.4open.science/r/figures-for-scones-561B

This repository includes figures showing the evolution of SCONES samples over the Langevin dynamics sampling process, as well as an expanded grid of qualitative samples for a variety of the problem settings considered in Table 1. Finally, there is a .pdf included which shows these figures as tables with captions.

---

### Decision · Program_Chairs · 2021-09-27

**Decision:**

Accept (Poster)

**Comment:**

This paper proposes a method to sample from high dimensional regularized optimal transport maps using Langevin diffusion on neural network parameterized potentials. The reviewers compliment the authors on being some of the first to tackle this problem at a truly large scale. The problem is important and the proposed solution seems technically sound and practically promising.

After the first round of reviews some concerns surfaced about the correctness of some of the presented results. In addition to general concerns about experimental validation from multiple reviewers, reviewer zJ3u pointed out some irregularities in the reported BW-UVP values which turned out to be caused by a scaling error in the regularization used in some of the experiments. The authors responded by providing updated results, in addition to acknowledging and fixing the error. The reviewers are satisfied with the author response and are now unanimous in their recommendation to accept the paper.

Authors, please carefully revise your paper for the camera ready version, and make sure to incorporate all new results and all changes that were promised in the discussion with the reviewers.